# Interactive Stratospheric Aerosol models response to different amount and altitude of SO$_2$ injections during the 1991 Pinatubo eruption

Ilaria Quaglia[1], Claudia Timmreck[2], Ulrike Niemeier[2], Daniele Visioni[3], Giovanni Pitari[1], Christina Brodowsky[4], Christoph Brühl[5], Sandip S. Dhomse[6], Henning Franke[2,7], Anton Laakso[8], Graham W. Mann[5,9], Eugene Rozanov[4,10,11], and Timofei Sukhodolov[10,11]

[1]Department of Physical and Chemical Sciences, Università dell'Aquila, 67100 L'Aquila, Italy
[2]Max Planck Institute for Meteorology, Bundesstr. 53, 20146 Hamburg, Germany
[3]Sibley School of Mechanical and Aerospace Engineering, Cornell University, Ithaca, NY, USA
[4]Institute for Atmospheric and Climate Science, ETH Zürich, Zürich, Switzerland
[5]Max Planck Institute for Chemistry, Mainz, Germany
[6]School of Earth and Environment, University of Leeds, Leeds, U.K
[7]International Max Planck Research School on Earth System Modelling, Bundesstr. 53, 20146 Hamburg, Germany
[8]Finnish Meteorological Institute, Atmospheric Research Centre of Eastern Finland, 70200 Kuopio, Finland
[9]UK National Centre for Atmospheric Science, University of Leeds, Leeds, UK
[10]Physikalisch-Meteorologisches Observatorium Davos and World Radiation Center, Davos, Switzerland
[11]St. Petersburg State University, St. Petersburg, Russia

**Correspondence:** Ilaria Quaglia (ilaria.quaglia@aquila.infn.it)

**Abstract.** A previous model intercomparison of the Tambora aerosol cloud has highlighted substantial differences among simulated volcanic aerosol properties in the pre-industrial stratosphere, and has led to questions about the applicability of global aerosol models for large magnitude explosive eruptions prior to the observational period. Here, we compare the evolution of

5 the stratospheric aerosol cloud following the well observed June 1991 Mt. Pinatubo eruption simulated with six interactive stratospheric aerosol microphysics models to a range of observational data sets.

Our primary focus is on the uncertainties regarding initial SO$_2$ emission following the Pinatubo eruption, as prescribed in the Historical Eruptions SO$_2$ Emission Assessment experiments (HErSEA), in the framework of the Interactive Stratospheric Aerosol Model Intercomparison Project (ISA-MIP). Six global models with interactive aerosol microphysics took part in this

study: ECHAM6-SALSA, EMAC, ECHAM5-HAM, SOCOL-AERv2, ULAQ-CCM and UM-UKCA. Model simulations are performed by varying the SO$_2$ injection amount (ranging between 5 and 10 Tg-S), and the altitude of injection (between 18-25 km).

The comparisons show that all models consistently demonstrate faster reduction from the peak in sulfate mass-burden in the tropical stratosphere. Most models also show a stronger transport towards the extratropics in the northern hemisphere, at

15 the expense of the observed tropical confinement, suggesting a much weaker subtropical barrier in all the models, that results in a shorter e-folding time compared to the observations. Furthermore, simulations in which more than 5 Tg-S of SO$_2$ are injected show an initial overestimation of the sulfate burden in the tropics and, in some models, in the northern hemisphere,

and a large surface area density a few months after the eruption compared to the values measured in the tropics and the in-situ measurements over Laramie. This draws attention to the importance of including processes such as the ash injection for the removal of the initial $SO_2$ and aerosol lofting through local heating.

## 1 Introduction

Large magnitude volcanic eruptions can emit sulfur dioxide ($SO_2$) and other gases directly into the stratosphere. An abrupt increase in stratospheric $SO_2$ creates a long-lived volcanic aerosol cloud that scatters incoming solar radiation, absorbs solar and infrared radiation, and affects the composition of the stratosphere. Such volcanic induced enhancements of the stratospheric aerosol layer exert strong direct effects on climate because they influence the Earth radiation budget and cool the surface via the reduced insolation (McCormick et al., 1995; Soden et al., 2002); they also show a range of indirect effects, due to the volcanic aerosols effects on stratospheric circulation, dynamics and chemistry (e.g., Robock et al., 2009; Timmreck et al., 2012; Kremser et al., 2016).

Here we investigate the evolution of the volcanic aerosol cloud after Mt. Pinatubo eruption in June 1991 by analysing coordinated simulations within the HErSEA (Historical Eruptions $SO_2$ Emission Assessment) experiments, in the framework of the Interactive Stratospheric Aerosol Model Intercomparison Project (ISA-MIP, Timmreck et al., 2018). Mount Pinatubo is located in the western part of the island of Luzon, Philippines (15.1°N, 120.4° E). After preliminary eruptions from 12 June 1991, the climatic phase started at 05:30 UTC on 15 June 1991 and lasted for approximately 9 hours. The volcanic cloud contained gases and particles of ice, ash, and sulfate, and reached a maximum altitude of 40 km (Holasek et al., 1996). Ice and ash burden peaked at about 80 and 50 Tg respectively, and early formed sulfate mass was estimated at 4 Tg, based on infrared satellite data from the Advanced Very High Resolution Radiometer and TIROS Operational Vertical Sounder/High Resolution Infrared Radiation Sounder/2 sensors (AVHRR, TOVS/HIRS/2;  Guo et al., 2004a). Initial sulfur dioxide ($SO_2$) mass estimates from the ultraviolet Total Ozone Mapping Spectrometer (TOMS) and infrared TOVS sensors, indicated that the eruption injected 14-22 Tg of $SO_2$ (Bluth et al., 1992; Guo et al., 2004a). Other uncertainties pertain to the vertical extension of the volcanic cloud: $SO_2$ mass was injected between 18-30 km (Bluth et al., 1992; Baran et al., 1993) and concentrated around 25 km, over a rich ash layer peaking around 22 km (Guo et al., 2004b). The sulfuric acid cloud peaked at 14 Tg in September (Lambert et al., 1993; Baran and Foot, 1994), with the largest aerosol concentration between 20 to 25 km of altitude and much lower amounts between 15 and 20 km (Winker and Osborn, 1992a, b; DeFoor et al., 1992). Recent volcanic $SO_2$ emission databases suggest for Pinatubo an amount and location of $SO_2$ emitted between 15 and 18 Tg of $SO_2$, at an altitude of between 19 and 28 km (Independent Volcanic Eruption Source Parameter Archive Version 1.0, ivespa.co.uk, VolcanEESM: Global volcanic sulphur dioxide (SO2) emissions database from 1850 to present - Version 1.0, Multi-Decadal Sulfur Dioxide Climatology from Satellite Instruments;  Aubry et al., 2021; Neely III and Schmidt, 2016; Carn, 2022).

Several modelling studies have evaluated the simulated global and tropical sulfate loadings compared to observations, with some studies (Niemeier et al., 2009; Toohey et al., 2011; Brühl et al., 2015) finding agreement when emitting in the mid-range of the best-estimate stratospheric $SO_2$ loading of 14-22 Tg $SO_2$ (Guo et al., 2004a). In contrast, a number of recent studies

found agreement only when injecting an amount of $SO_2$ below the lower limit of that observed, considering different injection heights and vertical distributions (Dhomse et al., 2014; Sheng et al., 2015a; Mills et al., 2016); this difference partly motivate the design of the ISA-MIP HErSEA intercomparison (see Timmreck et al., 2018). Approaching the problem from a model intercomparison perspective, different past projects have revealed large differences in the simulation of the aerosol radiative forcing, and not just for Pinatubo.

A first multi-model inter comparison study of global stratospheric interactive aerosol models was set up in the frame of the Model Intercomparison Project on the climatic response to Volcanic forcing (VolMIP, Zanchettin et al., 2016). To create a common forcing data set for the VolMIP volc-long-eq experiment, which considers a volcanic eruption with radiative forcing comparable to that of the 1815 Tambora eruption, a VolMIP pre-study was set up. This VolMIP-Tambora ISA experiment establishes a well defined set of injection parameters to simulate the Tambora volcanic aerosol cloud interactively with strato-spheric aerosol models. Multi-model analysis of the simulated volcanic aerosol distribution show large inter-model differences (Marshall et al., 2018; Clyne et al., 2021).

Marshall et al. (2018) used Arctic and Antarctic ice core information about sulfate deposition to constrain the VolMIP-Tambora ISA model simulations. The four models involved in this experiment revealed large discrepancies in the simulated aerosol burden, resulting in depositions magnitude in Antarctic ranging from 19 to 264 $kg\,km^{-2}$. They attributed the differences between the models, and between models and observations, to different sulfate formation and transport through meridional cir-culation and stratosphere-troposphere exchange, and different deposition schemes. The contribution to the overall uncertainty of the sulfate formation processes was then further investigated in a subsequent study by Clyne et al. (2021), which focused on the evolution of the global stratospheric aerosol optical depth. The reasons for the discrepancies between the models were attributed to differences in particle size, which influence the scattering efficiency and the lifetime of the stratospheric aerosols, and the treatment of hydroxyl radical (OH) chemistry, which in turn affects the timing of sulfate formation.

The Geoengineering Model Intercomparison Project Phase 6 (GeoMIP6, Kravitz et al., 2015) also includes experiments with injection of stratospheric sulfate aerosols precursors (G6Sulfur) in an amount necessary to reduce the net radiative forcing from the SSP5-8.5 scenario to the SSP2-4.5 one. Participating models in G6Sulfur directly injected $SO_2$ in the tropical stratosphere with different altitude and latitude ranges of injection or prescribed the aerosol optical depth or aerosol distribution derived from previous simulations. The amount of $SO_2$ required to achieve the proposed cooling varies by a factor of 2 between models, and results in a different temporal and latitudinal distribution of aerosols that affects surface temperature and local precipitation differently (Visioni et al., 2021).

In contrast to the aforementioned model intercomparison studies, the ISA-MIP HErSEA experiments offer a test of the relia-bility of these models by allowing a direct comparison of the simulated volcanic enhancement of the stratospheric aerosol layer with observation data sets, especially during the Mt. Pinatubo eruption, for which several satellite and in-situ measurements are available. Hence, HErSEA was developed to determine which set of volcanic emission source parameters allows models to reproduce the available measurements, and understand how their different chemical and microphysical schemes, stratospheric dynamics, and radiative transfer treatment influence these choices. Specifically, HErSEA focuses on the uncertainty in the initial volcanic emission in terms of amount and injection altitude of $SO_2$ for the recent large-magnitude volcanic eruptions

in the last 100 years (Mt. Agung 1963, Mt. El Chichón 1982, Mt. Pinatubo 1991); multiple interactive stratospheric aerosol simulations of each of the volcanic aerosol clouds with common upper-, mid- and lower-estimate amounts and injection altitudes of sulfur dioxide were performed. Here we investigate the evolution of the volcanic aerosol cloud after Mt. Pinatubo eruption by analysing Atmospheric Model Intercomparison Project (AMIP)-type (Gates et al., 1999) simulations within the HErSEA framework. In particular, we ask whether previous results in inter-model differences are confirmed in this new MIP; the presence of multiple injection settings common between all models will also allow an exploration of the reason for these differences, based on the models abilities to reproduce observations with different sets of initial conditions of the volcanic emissions.

The experimental design, the main features of the participating models and the observational data sets are described in Section 2. Section 3 shows model results of the optical and microphysical properties of the volcanic aerosol cloud, which are summarised and discussed in Section 4.

## 2 Methods and Data

### 2.1 Methods

#### 2.1.1 Experimental Protocol

There is a degree of uncertainty over the thickness of the injected $SO_2$ cloud, based on available measurements. Therefore, different modelling centers may have selected in the past different simulated injection altitudes for the Pinatubo eruption. Within (Dhomse et al., 2020) UM-UKCA set the $SO_2$ injection altitude at 21-23 km based on the altitude of the first detection of the Pinatubo cloud at Mauna Loa (Antuña et al., 2002). Further UM-UKCA analysis by Shallcross (2020) demonstrated improved model correspondence with the July-Aug 1991 Mauna Loa lidar measurements when running the model with "pre-nudged free-running", rather than the "approximate QBO free-running" approach used in (Dhomse et al., 2020). Sheng et al. (2015b) performed with AER 2-D 300 atmospheric simulations of the Pinatubo eruption by varying the emission parameters and found agreement with several observations by injecting 14 Tg of $SO_2$ with a vertical distribution peaking at 18-21 km. Similar emission parameters (10-12 Tg of $SO_2$ at 18-20 km) were used in Mills et al. (2016) with CESM1-WACCM. Niemeier et al. (2009) showed comparable aerosol optical depth and effective radius with satellite and lidar measurements, simulating with MAECHAM5-HAM the injection of 17 Tg of $SO_2$ at about 24 km together with 100 Tg of fine ash at about 21 km. Stenchikov et al. (2021) simulated with WRF-Chem v3.7.1 the same amounts of $SO_2$ and ash but centred at 17 km showing that the radiative heating of ash can raise the sulfur cloud by 7 km during the first week of the eruption. These differences motivated the design of the ISA-MIP HErSEA intercomparison.

The HErSEA Pinatubo experiment design includes five different emission scenarios considering different amounts and altitudes of injection of $SO_2$, as summarised in Figure 1. The first three emission scenarios describe injections at medium altitude (between 21-23 km) of an amount of $SO_2$ that varies from the lowest values of 5 Tg-S (Low-22km), to medium of 7 Tg-S (Med-22km), to the highest of 10 Tg-S (High-22km). The medium injection scenario (7 Tg-S of $SO_2$) has three different

injection altitude settings: Med-22km, as discussed, another shallow one at lower altitudes (18-20 km, Med-19km) and one over deep altitude-range (18–25 km, Med-18-25km).

The Mt. Pinatubo-like eruption is timed on June 15, 1991. $SO_2$ is injected in models in a single grid-cell close to the Pinatubo location (15°N, 120°E) and at the prescribed altitudes, with the precision given by the specific vertical and horizontal model resolution (table S1). UM-UKCA provided an additional set of simulations, called meridional-spread injection simulations, and EMAC simulation differ from the protocol: this differentiation is highlighted by the addition of a * after the model name. In UM-UKCA*, $SO_2$ is injected at Mt. Pinatubo longitude and in a latitude range between 0° and 15°N (12 model grid boxes),

a common strategy (Dhomse et al., 2014; Mills et al., 2016) to match the initial southward spread of the aerosol cloud (Bluth et al., 1992). In EMAC (we will use EMAC* only in the figures and tables), volcanic $SO_2$ injections are entered at one single point in time as 3D-mixing ratio perturbations derived from satellite data using an inventory for the period 1990 to 2019 (https://doi.org/10.26050/WDCC/SSIRC_3). For the Pinatubo period also the eruptions of Cerro Hudson (August 10, 1991), Spurr and Lascar are included in EMAC. The amount of $SO_2$ injected is 8.5 and 0.65 Tg-S for Pinatubo and Cerro Hudson,

respectively, and top heights of the volcanic plumes are approximately 23 km and 18 km.

All models are radiatively coupled to the volcanically enhanced stratospheric aerosol in order to resolve the composition–radiation–dynamics interactions. Previous model studies (e.g., Young et al., 1994; Timmreck et al., 1999; Aquila et al., 2012; Sukhodolov et al., 2018) showed that inclusion of the interaction between volcanic sulfate aerosol and radiation is essential for a reliable simulation of the transport of the volcanic cloud. Radiative heating of ash and $SO_2$ is also important for

the initial uplift of the volcanic cloud (Lary et al., 1994; Young et al., 1994; Gerstell et al., 1995), but the contribution of $SO_2$ is smaller than that of ash, in the first week, or sulfate aerosols, in the subsequent weeks (Stenchikov et al., 2021). About 80 Tg of ash was injected during the Pinatubo eruption (Guo et al., 2004b). However, both ash and $SO_2$ radiative effects are not included in all model simulations as it is outside the scope of the project which focuses on the long-term evolution of the Pinatubo volcanic cloud.

Modelling groups performed transient AMIP-type (Atmospheric Model Intercomparison Project) (Gates et al., 1999) runs of the Mt. Pinatubo eruption in which sea surface temperatures and sea ice extent are prescribed as monthly climatologies from the MetOffice Hadley Center Observational data set (Rayner et al., 2003). Boundary conditions are prescribed also for greenhouse gases and ozone depleting substances as recommended for the SPARC CCMI (Stratosphere-troposphere Processes And their Role in Climate Chemistry-Climate Model Initiative) hindcast scenario REFC1SD (Eyring et al., 2013), in order to match those

for the time period. The evolution of the quasi-biennal oscillation (QBO) must be consistent through the post-eruption period, as it affects the dispersion of the volcanic plume to mid-latitudes (Trepte and Hitchman, 1992; Baldwin et al., 2001; Punge et al., 2009), and consequently the size distribution and lifetime of stratospheric aerosols (Hommel et al., 2015; Pitari et al., 2016b; Visioni et al., 2017). Accordingly, models with internally generated QBO re-initialized it in order to be consistent with the actual meteorological conditions, or used specified dynamics approaches (e.g. Telford et al., 2008). All groups submitted

a 3-member ensemble for each different injection setting, except for ULAQ-CCM and EMAC, which submitted only one realization. The generation of the ensemble for each model is explained in the respective sections describing the model. Unless otherwise specified, all results shown refer to the ensemble mean.

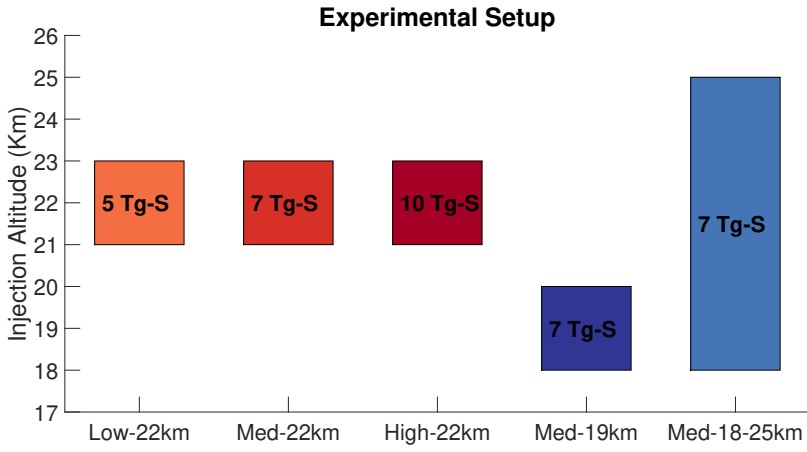

**Figure 1.** Graphical representation of injection setting parameters. The reddish boxes represent an injection of 5, 7, and 10 Tg-S of $SO_2$ centred at 22 km; the blue and light blue boxes represent the injection of 7 Tg-S of $SO_2$ for injection altitudes centred at 19 km, and one deep injection between 18 and 25 km.

#### 2.1.1.1 Cerro Hudson simulations

To evaluate the role of the Cerro Hudson eruption, we performed two additional simulations with the ULAQ-CCM model that, while outside the scope of ISA-MIP, helped clarify some issues raised by the initial results. The two simulations add the Cerro Hudson eruption to the Med-22km experiment with lower and upper estimates of $SO_2$ injection based on the Neely III and Schmidt (2016) and MSVOLSO2L4 inventory (Carn, 2022), respectively. The additional eruption consists in the injection of $SO_2$ with a uniform vertical distribution on August 10, 1991 in the grid-cell corresponding to the Cerro Hudson location (45.9°S, 72.9°W). The lower-end emission, termed Med-22km + Low-Hud, includes 1.5 Tg of $SO_2$ between between 11 and 16km, the upper-end emission Med-22km + High-Hud 4 Tg of $SO_2$ at 12-18km.

#### 2.1.2 Participating Models

The ISA-MIP multi-model ensemble includes simulations from five global aerosol models: ECHAM6-SALSA (Sect. 2.1.2.1), ECHAM5-HAM (Sect. 2.1.2.2), SOCOL-AERv2 (Sect. 2.1.2.3), ULAQ-CCM (Sect. 2.1.2.4), UM-UKCA (Sect. 2.1.2.5). In addition closely related simulations from a sixth model, EMAC, are considered (Sect. 2.1.2.6). The main characteristics of the participating models are reported in Table 1. ECHAM5-HAM, SOCOL-AERv2 and EMAC are based on the same general circulation model (GCM), ECHAM5 (Giorgetta et al., 2006), but with different horizontal and/or vertical resolutions, while ECHAM6-SALSA uses the updated version ECHAM6.3 (Stevens et al., 2013); all have different chemical and aerosol modules.

#### 2.1.2.1 ECHAM6-SALSA

ECHAM6-SALSA (ECHAM6.3-HAM2.3-MOZ1.0) is an interactive aerosol-chemistry-climate model based on ECHAM6.3 general circulation model (Stevens et al., 2013). A T63L95 resolution was used in ECHAM6-SALSA simulations, which corresponds to an approximately 1.9°x1.9° horizontal grid and 95 vertical layers reaching up to 80 km. The QBO is internally resolved by the model (Laakso et al., 2022). The GCM is interactively coupled with the HAMMOZ aerosol-chemistry model (Schultz et al., 2018) that is a combination of the Hamburg Aerosol Model (HAM) and the Model for OZone And Related chemical Tracers (MOZART) chemistry model. However MOZART was not used in the simulations for this study and OH and ozone concentrations were prescribed by a monthly mean climatology; a simplified sulfate chemistry scheme of HAM was used. The aerosol model HAM calculates the emissions, removal, and radiative properties of aerosol. It simulates five major global aerosol compounds: sulfate, organic carbon, black carbon, sea salt and mineral dust. The aerosol emissions from anthropogenic sources were based on the Community Emission Data System (CEDS) for CMIP6 anthropogenic emission inventory. Sea salt and dust emissions were calculated online. Aerosol microphysics were calculated by the sectional aerosol module SALSA. A detailed description of the Model is given in Kokkola et al. (2018). SALSA describes aerosols using 10 size bins in size space and the seven largest bins are separated into externally mixed soluble and insoluble populations. Ensemble members were produced by using insignificantly different values for one of the tuning parameters (the rate of snow formation by aggregation) for January 1991 of each ensemble member.

#### 2.1.2.2 ECHAM5-HAM

ECHAM5-HAM has the ECHAM5 GCM (Giorgetta et al., 2006), used as a high-top model in the middle atmosphere (MA) version, and interactively coupled to the aerosol microphysical model HAM (Stier et al., 2005). The horizontal resolution is about 2.8° in longitude and latitude, in a spectral truncation at wave number 42 (T42), with 90 vertical layers up to 0.01 hPa (about 80 km) and an interactive simulation of the QBO. The aerosol microphysical model HAM (Stier et al., 2005) calculates the oxidation of sulfur and sulfate aerosol formation, including nucleation, accumulation, condensation, and coagulation processes. The width of the HAM modes has been adapted to the conditions under a high sulfur load. The aerosols are prescribed in three modes with a fixed width (Niemeier et al., 2009). HAM was further adopted to stratospheric conditions by applying a simple stratospheric sulfur chemistry above the tropopause (Timmreck, 2001; Hommel et al., 2011). ECHAM prescribes oxidant fields of OH, $NO_2$, and $O_3$ on a monthly basis, as well as photolysis rates of OCS, $H_2SO_4$ $SO_2$, $SO_3$, and $O_3$. The sulfate was radiatively active for both SW and LW radiation and coupled to the radiation scheme of ECHAM. Further details are described in Niemeier et al. (2021). The ensemble members were produced by increasing the stratospheric horizontal diffusion from one level to the next above on January 1 of the year of the eruption. The parameter generating different dynamical state is perturbed between 1.0, 1.0001 and 1.001.

### 2.1.2.3 SOCOL-AERv2

SOCOL-AERv2 is an interactive aerosol-chemistry-climate model that is also based on the ECHAM5 GCM but coupled to the MEZON chemistry (Egorova et al., 2003) and AER sulfate aerosol microphysics (Weisenstein et al., 1997) modules. The model version used here has a horizontal resolution of about 2.8° in longitude and latitude (T42) and 39 vertical layers up to 0.01 hPa. Because of the coarse vertical resolution (~1.5 km in the lower stratosphere), the QBO is nudged to the observed equatorial wind profiles. The chemistry module calculates the interactions of 89 chemical species of the oxygen, hydrogen,

nitrogen, carbon, chlorine, bromine, and sulfur groups in gas-phase, photolysis, and heterogeneous reactions, including reactions in/on aqueous sulfuric acid aerosols. The sulfate aerosol module resolves the aerosol particles in 40 size bins (the highest aerosol size resolution compared to other participating models), ranging in dry radius from 0.39 nm to 3.2 $\mu$m, and calculates nucleation, condensational growth, evaporation, coagulation, and sedimentation of sulfate aerosol bins. $H_2SO_4$ weight percent is calculated online based on actual temperature and relative humidity. Dry and wet deposition of species are interactively

calculated based on actual meteorological conditions in the model (Feinberg et al., 2019). Modelled aerosols and chemical species are coupled with the short-wave and long-wave radiation schemes. Aerosol radiative properties are treated following a lookup-table approach with precalculated values using Mie theory for actual $H_2SO_4$ weight percent and temperature. All boundary conditions follow the recommendations of ISA-MIP (Timmreck et al., 2018). Three ensemble members were produced by scaling the global $CO_2$ concentration by ±0.05%, which started in January 1991 and was maintained for the whole

simulation. Besides the 39-level version, SOCOL-AERv2 can also be run on 90 levels, as the other two ECHAM5-based participating models ECHAM5-HAM and EMAC. However, increased resolution more than doubles the computational expenses of the already heavy calculations of interactive chemistry and highly resolved sectional aerosol microphysics. Therefore, the model is mostly used in the 39-level configuration. To test the effects of increased resolution, SOCOL-AERv2 has been additionally used here for the Low-22km experiment with the 90 levels instead of the reference 39. With this configuration, the

model has been spun up to the conditions of 1991. Besides changed resolution, all other setting have been kept the same.

### 2.1.2.4 ULAQ-CCM

ULAQ-CCM (University of L'Aquila Chemistry Climate Model) is a global scale climate-chemistry coupled model with a horizontal resolution of 5°x6° (T21) and 126 log pressure levels (approximate pressure altitude increment of 568 m), from the surface to the mesosphere (0.04 hPa). However, the QBO is not internally resolved and is nudged to observed values (Morgen-

225 stern et al., 2017) and its future values are repeated from the historical time series. The chemistry module includes medium and short-lived species ($O_x$, $NO_y$, $NO_x$, $CHO_x$, $Cl_y$, $Br_y$, $SO_x$) and the major component of stratospheric and tropospheric aerosols (sulfate, nitrate, organic and black carbon, soil dust, sea salt, polar stratospheric clouds). The microphysical code for aerosol formation and growth includes gas-particle conversion scheme, homogeneous and heterogeneous nucleation, coagulation, condensation and evaporation (Pitari et al., 2002, 2016a). It also includes heterogeneous chemical reactions on sulfuric

acid aerosols and polar stratospheric cloud particles; both heterogeneous and homogeneous upper tropospheric formation processes are also included (Visioni et al., 2018a). The aerosol module calculates the aerosol extinction, asymmetry factor, and

single scattering albedo, given the calculated size distribution of the particles for different wavelength and they are passed daily to the radiative transfer module that is is a two-stream delta-Eddington approximation model (Toon et al., 1989).

#### 2.1.2.5 UM-UKCA

UM-UKCA model simulations are performed using Global Atmosphere 4.0 configuration (Walters et al., 2014, GA4) of the UK Met Office Unified Model (UM v8.4) general circulation model with the UK Chemistry and Aerosol chemistry–aerosol sub-model (UKCA). The GA4 atmosphere model has a horizontal resolution of 1.875°×1.25° and 85 vertical levels (N96L85) ranging from the surface to about 85 km, with an interactive simulation of the QBO. UM-UKCA configuration adapts GA4 with aerosol radiative effects from the interactive GLOMAP aerosol microphysics scheme and ozone radiative effects from the whole-atmosphere chemistry that is a combination of the detailed stratospheric chemistry and simplified tropospheric chemistry schemes (Archibald et al., 2020). GLOMAP stratospheric aerosol microphysics scheme is described in Dhomse et al. (2014), and model setup is described in Dhomse et al. (2020). Briefly, the model uses the GLOMAP aerosol microphysics module coupled with troposphere-stratosphere chemistry scheme and modelled aerosols are coupled with the radiation scheme. Model also uses Greenhouse gas (GHG) and ozone-depleting substance (ODS) concentrations from Ref-C1 scenario used in the CCMI-1 (Morgenstern et al., 2017) activity. Simulations are performed in atmosphere-only mode, and CMIP6 recommended sea-surface temperatures and sea-ice concentration that are obtained from https://esgf-node.llnl.gov/projects/cmip6/ (last access: 25 March 2021) are used. Three ensemble members were initialised using the fields of three model years of a 20-year time-slice simulations prior 1990 that gave a QBO transition approximately matching that of ERA-Interim reanalysis (Dee et al., 2011; Dhomse et al., 2020, for more details).

#### 2.1.2.6 EMAC

EMAC is the ECHAM5 general circulation model coupled with the Modular Earth Submodel System Atmospheric Chemistry (Brühl et al., 2015, 2018). The resolution is T63/L90, i.e. about 1.9° latitude and longitude and 90 layers up to about 80 km with a vertical resolution of about 500 m near the tropopause. The QBO is internally generated but slightly nudged to observations compiled by the Free University of Berlin. Below 100 hPa and above the boundary layer dynamics and temperature are nudged to ERA-Interim. It contains comprehensive gas-phase and heterogeneous chemistry. The applied aerosol module GMXE (Pringle et al., 2010) accounts for seven modes using lognormal size distributions (nucleation mode, soluble and insoluble Aitken, accumulation and coarse modes). The boundary between accumulation mode and coarse mode, a model parameter, is set at a dry particle radius of 1.6 $\mu$m to avoid too fast sedimentation of a too large coarse mode fraction in case of major volcanic eruptions. Optical properties for the types sulfate, dust, organic carbon and black carbon (OC and BC), sea salt, and aerosol water are calculated using Mie-theory-based lookup tables for each mode consistent with the selected size distribution widths of the modes. This also means that no overall effective radius is used. The resulting total optical depths, single scattering albedos and asymmetry factors are used in radiative transfer calculations which feedback to atmospheric

**Table 1.** Main chemical, microphysical and dynamic characteristics of the participating models. * highlight models with spatially spread SO$_2$ injections.

| Model | Injection region | Interactive OH | Stratospheric aerosol components | Aerosol dynamics scheme | Simulated aerosol in het. chem. | Nucleation scheme | QBO |
|---|---|---|---|---|---|---|---|
| ECHAM6-SALSA | Point | N | Sulfate, Dust, OC, BC and SS | 2-moment sectional, 10 bins | N | Vehkamäki et al. (2002) | Internally generated |
| ECHAM5-HAM | Point | N | Sulfate | 2-moment modal, 7 modes | N | Vehkamäki et al. (2002) | Internally generated |
| EMAC | 3D-plume | Y | Sulfate, Dust, OC, BC, aerosol water | Modal, 7 modes | Y | Vehkamäki et al. (2002) | Internally generated but slightly nudged |
| SOCOL-AERv2 | Point | Y | Sulfate | Sectional, 40 size bins | Y | Vehkamäki et al. (2002) | Nudged |
| ULAQ-CCM | Point | Y | Sulfate (also other components in troposphere) | Sectional, 22 bins | Y | Pitari et al. (1993) | Nudged |
| UM-UKCA* | 0-15°N, 120°E | Y | Sulfate and Meteoric Smoke particles | 2-moment modal, 7 modes | N | Mann et al. (2010) | Internally generated |

dynamics. The results from EMAC were taken from an existing 30 years transient simulation for comparison (Schallock et al., 2021).

## 2.2 Observation data sets

### 2.2.1 AVHRR

The Advanced Very High Resolution Radiometer (AVHRR/2) is a space-borne sensor that measures the reflectance of the Earth in five spectral bands covering visible and infrared wavelengths (0.63, 0.86, 3.7, 11, 12 $\mu$m). AVHRR/2 instrument was on board of the polar-orbiting satellites (POES) NOAA-11 that provided global coverage data with a resolution of 1.1 km and a frequency of earth scans twice per day (https://www.avl.class.noaa.gov/release/data_available/avhrr/index.htm). The data used here are on a 1°x1° grid as monthly averages (as archived at NOAA's National ClimateData Center). As in Long and Stowe (1994) and Aquila et al. (2012), the stratospheric optical depth at 0.5 $\mu$m is calculated by removing monthly mean background values (June 1989 to May 1991) from AVHRR observations. The optical depth at 0.5 $\mu$m is retrieved through a radiative transfer surface/atmosphere model (RAO et al., 1989) therefore, combined with the previous assumption, AVHRR can not detect the changes of stratospheric AOD smaller than 0.01 but can detect values up to 2.0 (Russell et al., 1996).

### 2.2.2 SAGE II

The Stratospheric Aerosol and Gas Experiment II (SAGE II) is a satellite-based sun photometer that was launched in October 1984 aboard the Earth Radiation Budget Satellite (ERBS) and retired in August 2005. The instrument measures the extinction of the solar radiation through the limb of the Earth's atmosphere in 7 channels ranging from 385 to 1020 nm, with a global coverage from 80°S to 80°N latitude and a vertical resolution of 1 km for the retrieved data (Mauldin et al., 1985). We used the effective radius and the surface area density of aerosol particles from SAGE II version 7.0 (Damadeo et al., 2013; NASA/LARC/SD/ASDC, 2012b). The SAD (and thus the effective radius) is derived by a method that is a linear mix between the Thomason et al. (1997) method, which is valid for the 525-1020 nm extinction ratio below 1.5, and the Thomason and Burton (2008) method for ratios above 2.0 (Damadeo et al., 2013). Both methods assume that aerosols are spherical droplets of H$_2$SO$_4$-H$_2$0 solution with a constant composition of 75% H$_2$SO$_4$ and 25% H$_2$0 by weight. The Thomason et al. (1997) method uses the principal component analysis to derive the SAD from a linear combination of four aerosol extinction measurements

(386, 452, 525, 1020 nm). In the Thomason and Burton (2008) method, SAD is derived from the 525 and 1020 nm channels using an empirical parameterization based on the 525-1020 nm extinction ratio.

The stratospheric sulfate burden is taken from the SAGE-3$\lambda$ data set (ftp://iacftp.ethz.ch/pub_read/luo/CMIP6/) that was compiled for phase 6 of the Coupled Model Intercomparison Project (CMIP6). $H_2SO_4$ density (and other secondary products not used here) is derived via the SAGE-3$\lambda$ algorithms that assume a single mode lognormal size distribution of stratospheric aerosol where number density, mode radius and width are obtained by fitting the SAGE II extinction coefficients at 3 wavelength (452, 525 and 1024 nm) (Revell et al., 2017).

### 2.2.3 HIRS

The High Resolution Infrared Radiation Sounder (HIRS) is an infrared scanning radiometer that has been onboard of several NOAA platforms starting with the first satellite of the Television Infrared Observation Satellite series (TIROS-N), followed by NOAA-6 up to NOAA-19 (Borbas and Menzel, 2021). It measures the reflectance of the earth in 19 infrared channels (3.7 to 15 $\mu$m) and one solar channel (0.69 $\mu$m) with a spatial resolution at nadir of 20.4 km on HIRS/2. Baran and Foot (1994) used HIRS/2 cloud-cleared radiances at 8.3 $\mu$m (NOAA-10/12) and 12.5 $\mu$m (NOAA-11) to retrieve the column density of sulfuric acid aerosols from May 1991 to November 1993. Among the assumption and the approximations, the stratospheric aerosols are assumed of 75% $H_2SO_4$ and 25% $H_2O$, with a spectral transmittance based on dustsonde measurements by Deshler et al. (1992) and a *single-scattering albedo calculated from Mie theory by integrating the extinction and scattering coefficients over a lognormal size distribution using a mode radius 0.35 $\mu$m and a normalized standard deviation of 1.6* (Baran and Foot, 1994). The data cover the latitudes from 80°N to 80°S and all longitudes with 5° of resolution and are affected by a systematic error of 10% due to the sensitivity of the retrieved method and uncertainties in the background.

### 2.2.4 OPC

The University of Wyoming balloon-borne Optical Particle Counter (OPC) is a spectrometer that measures the light-extinction cross section of the particles using a broadband incandescent light source, developed by Rosen (1964), providing the particle size and the number concentration. The stratospheric aerosol measurements from 1991 to 2012 are made over Laramie (Wyoming) with the so-called OPC40, that can detect particles throughout the size range 0.1-10.0 $\mu$m, distinguished in 8 or 12 channels, depending on the instruments (Deshler, 2003). Here we used the revised data set (UWv2.0, http://www.atmos.uwyo.edu/ deshler/D of the OPC measurements (Deshler et al., 2019). Surface area density and volume density are calculated from the size distribution derived from particle size and concentration by fitting the data to a unimodal or bimodal lognormal distribution (depending on the number of measurements and of which of the two minimizes the difference between the calculated and the measured number concentration) (Kovilakam and Deshler, 2015).

### 2.2.5 GloSSAC

The Global Space-based Stratospheric Aerosol Climatology (GloSSAC) is a global and gap-free data set of zonally averaged optical properties of stratospheric aerosols (focused on aerosol extinction coefficient at 525 and 1020 nm) from 1976-2018. It is mainly based on the Aerosol and Gas Experiment (SAGE), and on the Optical Spectrograph and InfraRed Imager System (OSIRIS) and the Cloud-Aerosol Lidar and Infrared Pathfinder Satellite Observation (CALIPSO). Ground, airborne and balloon-based instruments were used to fill major gaps in the data set (Thomason et al., 2018). Here, we used the updated version v2 (NASA/LARC/SD/ASDC, 2012a) from Kovilakam et al. (2020).

## 3 Results

The various sets of initial conditions of SO$_2$ injections result in an aerosol cloud with different optical properties depending on the dispersion of the cloud over time and the size of the aerosols produced.

In the following section, we start by analysing the aerosol optical depth (AOD) and how the models reproduce the measured AOD with different volcanic emission source parameters. Since the amount of attenuation depends on the particle number concentrations and size, we then investigated both the magnitude and distribution of the sulfate burden and the size of the sulfate aerosols.

### 3.1 Aerosol optical depth

The stratospheric AOD simulated by the different interactive aerosol microphysical models is evaluated by comparing it with satellite observations from AVHRR and GloSSAC (Fig. 2). The AOD is calculated at a wavelength of 550 nm in EMAC, ECHAM5-HAM, ULAQ-CCM and UM-UKCA, 533 nm in ECHAM6-SALSA, 525 nm in SOCOL-AERv2 and GloSSAC, and 600 nm in AVHRR; differences between those wavelengths are however negligible. GloSSAC provides zonal values with a latitudinal resolution of 5° and uniform spatio-temporal coverage up to the year 1994. As it is mostly based on SAGE II measurements, the instrument saturates for optical depth of about 0.15, therefore it is less accurate in the centre of tropical cloud in the first months after the eruption (Russell et al., 1996). Conversely, AVHRR can only measure stratospheric AOD larger than 0.01. Because of the paucity of data points, "global values" when comparing against AVHRR are calculated between 60°S-60°N.

Figure 2 shows the time evolution of the zonal mean stratospheric AOD for each model and ensemble mean. It is clear that medium and high injection of SO$_2$ (Med-22km and High-22km, respectively) overestimate the stratospheric AOD in the tropics or/and in the Northern Hemisphere (NH) extratropics compared to both observations. The ability to reproduce the observed values in the Southern Hemisphere (SH) extratropics depends both on the model and the injection parameters. UM-UKCA* and EMAC, contrary to other models, show more southward transport, probably due to the different injection settings (see section 2.1.1). In UM-UKCA* the meridional-spread emission (0-15°N) accounts for the initial west-southwestward drift of the volcanic cloud (Bluth et al., 1992), contributing to a more hemispherically symmetric aerosol distribution (Dhomse et al.,

2014; Mills et al., 2016; Jones et al., 2017). EMAC used a 3D-plume injection and also included smaller eruptions such as that of Cerro Hudson in the southern hemisphere in August 1991 (45.9°S, 72.9°W). The additional injection is a 3D-plume injection of 0.65 Tg-S of $SO_2$, whose maximum in terms of mixing ratio is at 18 km, and differs from the two additional cases performed with ULAQ-CCM (2.1.1.1). In ULAQ-CCM, the Med-22km+Low-Hud includes a similar amount of $SO_2$ but at lower altitudes compared to the Cerro Hudson eruption in EMAC, and its effect on the stratospheric burden and AOD is negligible. In contrast, Med-22km+High-Hud enhances them in the southern hemisphere, approaching observation, but only for a few months after the eruption (Fig. S6).

A quantitative comparison with the observations is shown with the use of Taylor diagrams (see Appendix A) in Figure 3. Model results are compared for the first year after the eruption with both AVHRR and GloSSAC (first row and second row, respectively) and for the second year only with GloSSAC (third row). Three-member ensembles, when provided, are represented with smaller circles of the same colour with respect to the ensemble mean of a specific simulation. In ECHAM6-SALSA, the differences between members of the same scenario are greater than those between scenarios because of differences in local winds at the time of the eruption in each ensemble-member. The impact of local winds is weaker when $SO_2$ is injected over the deep altitude-range between 19 and 25 km (blues circles in Figure 3 panels a and h). There are various sets of initial conditions for $SO_2$ injections which, depending on the model, are close to the observations. The experiments that best reproduce the observations are those with similar variability to that of the observations, defined by their standard deviations (STDs), higher correlation (COR) and lower root-mean-square difference (RMSD). The values of COR and RMSD for these experiments are summarised in Table 2.

During the first year after the eruption, all models show better agreement with AVHRR than GloSSAC: correlations range between 0.73 and 0.78 with AVHRR versus 0.54 e 0.82 with GloSSAC, for which RMSDs are also higher. In ECHAM6-SALSA, SOCOL-AERv2 and ULAQ-CCM, the injection of 7 Tg-S of $SO_2$ closer to the tropopause is a good compromise between the too high and too low stratospheric AOD produced in the tropics by an injection of 5 and 10 Tg-S of $SO_2$, respectively, and this scenario produces also a better southward and northward transport (Fig. 2). The best set of initial parameters also depend on the observation considered for comparison: in ECHAM6-SALSA Med-18-25km and Med-19km reproduce better AVHRR and GloSSAC measurements, respectively, and in the comparison with GloSSAC the correlation increases and an RMSD decreases over time (Fig. 2 panel a5). For SOCOL-AERv2 and ULAQ-CCM, Med-19km is in good agreement with both AVHRR and GloSSAC in the two different period considered (Fig. 2 panels c4 and d4). During the first year after the eruption, the correlation between Med-19km and the observations is higher for ULAQ-CCM (0.84 and 0.74 compared with AVHRR and GloSSAC, respectively) as it better reproduce the tropical confinement, while the following year (June 1992 - July 1993), in SOCOL-AERv2 comparable values of SAOD persist for longer in the extratropics compared with GloSSAC (correlation of 0.86). In ECHAM5-HAM the injection at 21-23 km results in a comparable stratospheric AOD in the tropics and SH extratropics compared to both observations, but overestimates Northern Hemispheric (NH) extratropics values by up to a factor of two (Fig. 2 panels b1, b2 and b3). The amount of $SO_2$ to obtain the highest correlation between modeling experiments and observations depends on the observation and on the period considered: High-22km and Low-22km when compared with AVHRR and GloSSAC during the first year after the eruption, respectively, Med-22km when compared with GloSSAC

**Table 2.** Correlation (COR) and root-mean-square-difference (RMSD) of the stratospheric AOD calculated between observations and model results, for the experiments that best reproduce the observations. The * highlights models with spatially distributed SO$_2$ injections.

| Model | AVHRR (June 91 - May 92) | | | GloSSAC (June 91 - May 92) | | | GloSSAC (June 92 - May 93) | | |
|---|---|---|---|---|---|---|---|---|---|
| | Experiment | COR | RMDS | Experiment | COR | RMDS | Experiment | COR | RMDS |
| ECHAM6-SALSA | Med-18-25km | 0.74 | 0.08 | Med-19km | 0.60 | 0.07 | Med-19km | 0.79 | 0.02 |
| ECHAM5-HAM | High-22km | 0.74 | 0.09 | Low-22km | 0.71 | 0.07 | Med-22km | 0.82 | 0.02 |
| EMAC | | 0.79 | 0.07 | | 0.54 | 0.10 | | 0.63 | 0.03 |
| SOCOL-AERv2 | Med-19km | 0.73 | 0.08 | Med-19km/Low-22km | 0.61 | 0.09 | Med-19km | 0.86 | 0.02 |
| ULAQ-CCM | Med-19km | 0.84 | 0.07 | Med-19km | 0.74 | 0.07 | Med-19km | 0.69 | 0.03 |
| UM-UKCA | Low-22km | 0.56 | 0.12 | Low-22km | 0.63 | 0.11 | Med-19km | 0.47 | 0.05 |
| UM-UKCA* | Low-22km | 0.87 | 0.07 | Low-22km | 0.82 | 0.09 | Med-19km | 0.86 | 0.02 |

the following year. In UM-UKCA, the point injection and meridional-spread emission agree that Low-22km better reproduces the stratospheric AOD of both observations during the first year after the eruption, as it shows a good tropical confinement and comparable values in the NH, and for the meridional-spread emission also in the SH (Fig. 2 panels e1 and f1). Therefore, the correlation is higher and the RMSD is lower for the meridional-spread emission experiment. The poleward transport, especially in the NH, is enhanced in Med-19km (Fig. 2 panels e4 and f4) and found to have a higher correlation with GloSSAC one year after the eruption (COR of 0.86 and 0.47 for UM-UKCA* and UM-UKCA, respectively). During the first year after the eruption, EMAC has comparable values in the tropics and northern mid-latitudes with respect to AVHRR, while in the southern mid-latitude the stratospheric AOD is up to twice as larger, and results in a correlation of 0.79. The correlation decreases to 0.63 when comparing with GloSSAC during the following year because of the more rapid decline of the stratospheric volcanic cloud.

The persistence of the volcanic aerosol in the stratosphere is shown in Figure 4, which represents the global normalised stratospheric optical depth. The Med-19km experiment is shown for all models, as it is the experiment which best reproduces the GloSSAC observations after June 1992 for all models, with the exception of Med-22km for ECHAM5-HAM and EMAC with the only experiment provided. The e-folding time, calculated as the time between the maximum and the 1/e value, is 13 months in AVHRR and 15 months in GloSSAC. This range includes ULAQ-CCM and UM-UKCA with an e-folding time of 14 months and UM-UKCA* of 15 months. Lower values were found for SOCOL-AERv2 with 12 months, ECHAM6-SALSA and ECHAM5-HAM with 11 months, and EMAC with 10 months.

### 3.2 Sulfate Burden

Figure 5 shows the time evolution of the global and tropical stratospheric sulfate burden of different injection set-ups for each model. The results of each model are compared with satellite measurements from HIRS and the SAGE-3$\lambda$ data set. Large

# Stratospheric AOD

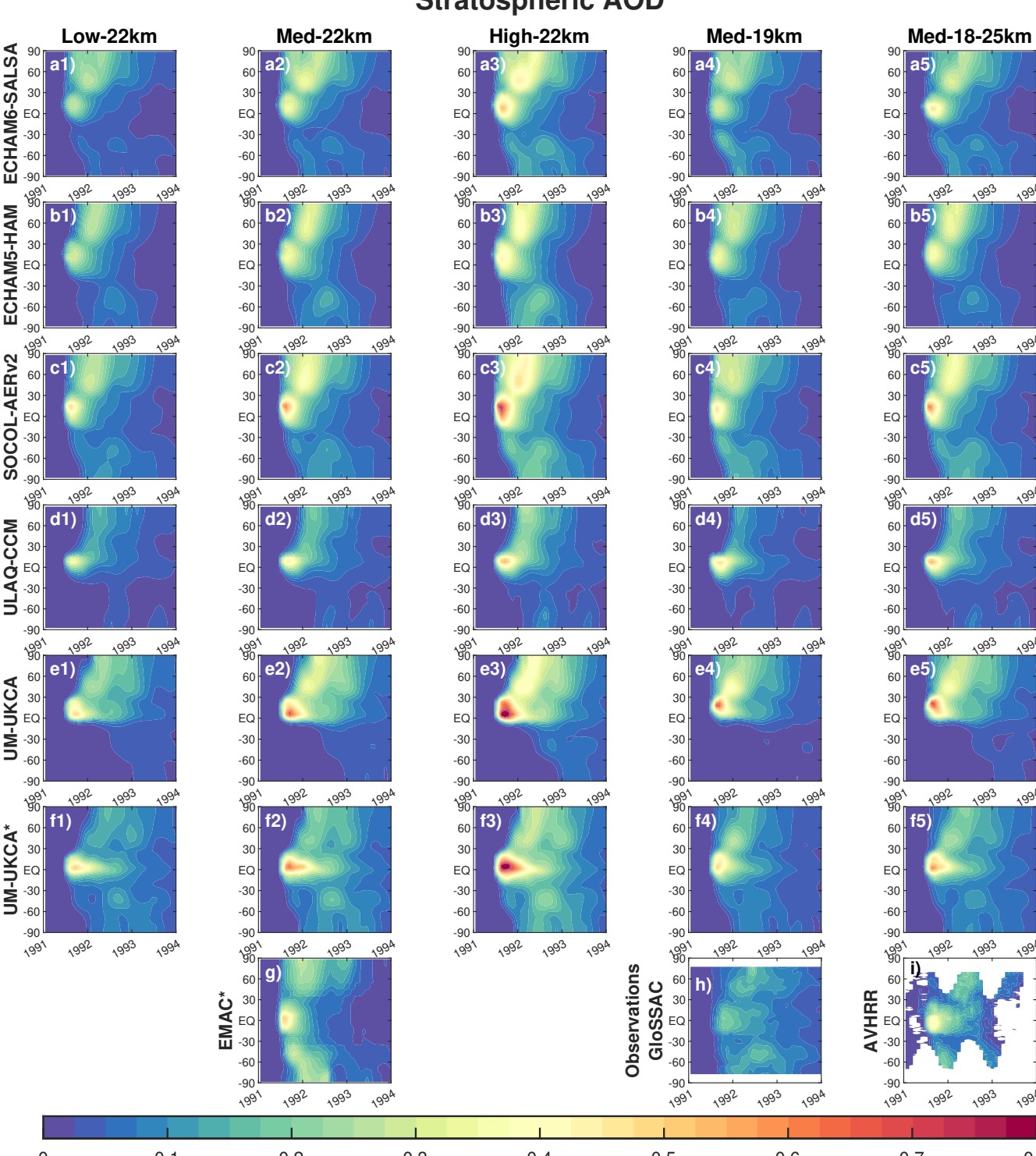

**Figure 2.** Time evolution of zonal stratospheric AOD for all models, in Low-22km (first column), Med-22km (second column), High-22km (third column), Med-19km (fourth column), Med-18-25km (fifth column). The last row includes the different scenario simulated by EMAC* and the two observations used for comparison: GloSSAC and AVHRR. AOD is calculated at a wavelength of 550 in ECHAM5-HAM, EMAC, ULAQ-CCM and UM-UKCA, 533 nm in ECHAM6-SALSA, 525 nm in SOCOL-AERv2, 525 nm in GloSSAC, 600 nm in AVHRR. * highlight models with spatially spread SO₂ injections.

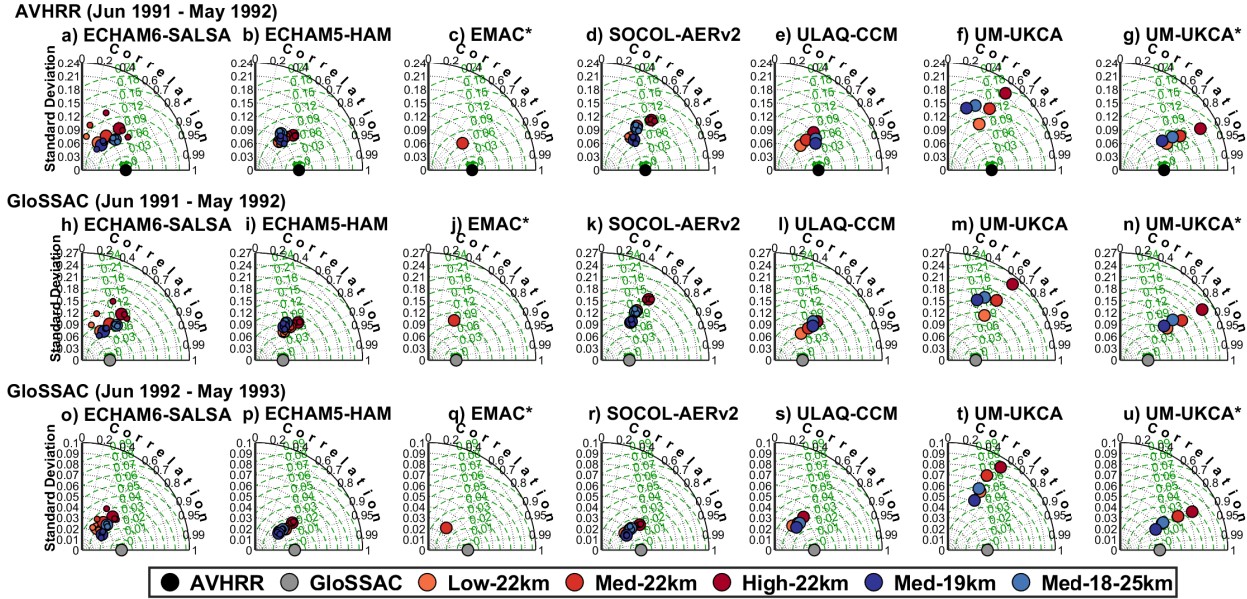

**Figure 3.** Taylor diagrams for the global stratospheric AOD. Zonal monthly mean values for different time periods have been used to calculate the standard deviation, correlation, centred root mean square difference between model experiments and measurements. In the 1st row, model results are compared with respect to AVHRR over the period June 1991 to May 1992; in the second row with respect to GloSSAC over the period June 1991 to May 1992; and in the third row with respect to GloSSAC over the period June 1992 to May 1993 (See appendix A1 for more details). * highlight models with spatially spread SO$_2$ injections.

differences are evident in the temporal evolution of the sulfate burden between the aerosol model simulation on one hand and the satellite data set on the other, which show similar values and a similar temporal evolution for the sulfate burden.

In the six months following the eruption (July-December, termed the build-up phase), ECHAM6-SALSA, ECHAM5-HAM, SOCOL-AERv2 and ULAQ-CCM best match the global stratospheric sulfate burden of HIRS and SAGE-3$\lambda$ with the injection 5 Tg-S of SO$_2$ (Low-22km), a lower amount compared to the one required for a comparable stratospheric aerosol optical depth (Fig. 5 panels a,b,d and e). For SOCOL-AERv2, Med-19km also shows values within the uncertainties of the HIRS measurements. However, Low-22km, and also Med-19km for SOCOL-AERv2, anticipates the peak and underestimates the tropical burden in ECHAM6-SALSA, ECHAM5-HAM and SOCOL-AERv2, while the peak is reached later and larger values are produced in ULAQ-CCM (Fig. 5 panels h,i,k and l). In UM-UKCA, point and meridional-spread injection show similar results for the global stratospheric sulfate burden and agree with observations with Med-19km and Med-18-25km experiments (Fig. 5 panels f and g). The differences between the two strategies emerge in the tropics where values are lower for point injection experiments due to the lack of aerosols transported to the southern tropics and therefore confined to the northern hemisphere. For the point injection, Low-22km and Med-18-25km approaches SAGE-3$\lambda$ for the first months and HIRS for the last 3 months of the build-up phase. All the experiments with larger amounts of injected SO$_2$, including the EMAC experiment

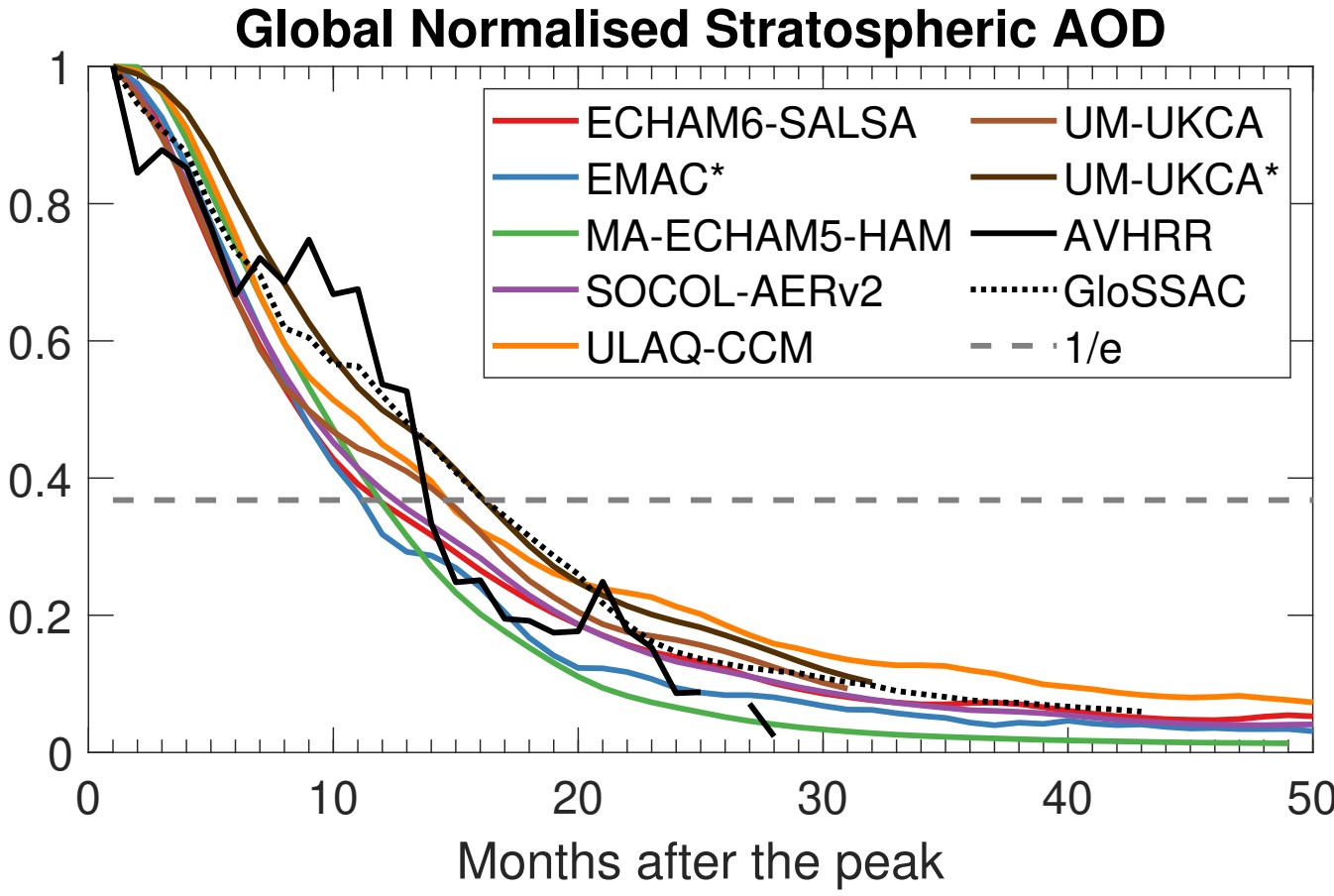

**Figure 4.** Time evolution of monthly values of the normalised global stratospheric AOD for models (colored lines) and AVHRR and GloS-SAC observations (black lines). The dashed gray line represents the 1/e value. The experiments shown are Med-19km for ECHAM6-SALSA, SOCOL-AERv2, ULAQ-CCM, UM-UKCA and UM-UKCA* and Med-22km for ECHAM5-HAM. For EMAC*, it refers to the only experiment provided. * highlight models with spatially spread $SO_2$ injections.

with 8.5 Tg-S of $SO_2$, overestimate the measured global sulfate burden; all experiments in ULAQ-CCM and the single scenario in EMAC overestimate the tropical burden, while in ECHAM6-SALSA, ECHAM5-HAM and SOCOL-AERv2 overestimate the burden in the NH extratropics (Fig. S5).

In the build-up phase, SAGE-$3\lambda$ assumes the lowest values and slowly reaches a peak of 5.0 Tg-S in December, compared to 5.4 Tg-S of HIRS in September. Lower values in SAGE-$3\lambda$ are related to the saturation effects of the limb-occultation instrument, therefore HIRS measurements are to be considered more reliable for this initial period (Sukhodolov et al., 2018). For EMAC, the injection of 8.5 Tg-S of $SO_2$ produces a sulfate aerosol cloud that peaks in September at 7.0 Tg-S, a value comparable to the results of the Med-22km experiment (performed by the other models), in which 7 Tg-S of $SO_2$ is injected. For SOCOL-AERv2 and UM-UKCA with both injection strategy, Med-19km shows the best agreement with HIRS in terms of peak and timing of the peak (September for SOCOL-AER, October for UM-UKCA), whereas in Low-22km and the other experiments it is reached one month later. This is followed by ECHAM6-SALSA in October (November only in High-22km) and ULAQ-CCM in November. ECHAM5-HAM is more sensitive to the altitude of injection: it peaks between October in Med-19km, November in Med-18-25km and December in the experiments with the same altitude of injection (Low-21km, Med-21km and High-21km); the values of the peak is 14.3% lower in Med-19 km and 7.1% lower in Med-18-25km compared to Med-22km.

The sensitivity to injection altitude depends on the model: during the build-up phase, the Med-18-25km and Med-22km curves coincide in ECHAM6-SALSA and SOCOL-AERv2, and, compared to these experiments, the values in Med-19km are up to 9% and 20% smaller for each model, respectively. In ULAQ-CCM, ECHAM5-HAM and UM-UKCA, the more $SO_2$ is injected at lower altitudes the smaller is the value of the peak but for ULAQ-CCM the peak is only 1% and 6% lower in Med-18-25km and Med-19km compared to Med-22km. Value and time of the peak for all models and experiments are summarised in table S2. In general, when the amount of $SO_2$ injected is exclusively or even in the lowest levels (Med-19km and Med-18-25km respectively), the sulfate burden is lower, and therefore this effect is less pronounced at Med-18-25km, as the aerosol distribution is more dependent on the balance between gravitational sedimentation in the lower stratosphere and the strength of vertical transport by the Brewer-Dobson Circulation, as well as the height of the tropopause.

Differences among models and experiments in terms of amount and timing during the build-up phase are influenced by the oxidation of $SO_2$ by OH that determines the timescale for aerosol formation (Clyne et al., 2021). For this reason, we distinguish between models with prescribed OH (ECHAM6-SALSA and ECHAM5-HAM) and those with interactive OH (SOCOL-AERv2, ULAQ-CCM, UM-UKCA) when looking at the $SO_2$ evolution. The global normalised $SO_2$ burden curves (fig. S4a) coincide for all models with prescribed OH. An exception of Med-19km in ECHAM6-SALSA, which has lower values and might depend on an early removal through tropopause flux, facilitated by injection near the tropopause. In ULAQ-CCM and UM-UKCA, when comparing High-22km with Low-22km we find that a higher injected $SO_2$ mass produces a longer initial e-folding time for $SO_2$. The same applies when comparing injections concentrated in a few kilometres (Med-22km and Med-19km), i.e. where $SO_2$ oxidation depletes OH more quickly (Mills et al., 2017), with those where the same amount of $SO_2$ is injected over a wider altitude band. Consequently, initial values of the stratospheric sulfate burden in Med-18-25km are slightly higher compared to Med-22km and Med-19km.

In order to better understand the models sensitivity to the different emission scenarios and eventual non-linearities, in Figure 6 we normalise the resulting global sulfate burden by the amount of $SO_2$ injected. Thus, in the build-up phase we would expect all the curves for all experiments to reach a value of 1, since no $SO_2$ and sulfate aerosols have yet been removed from the atmosphere. This will highlight the differences in the aerosol removal (wet removal, deposition, sedimentation) depending on the injection altitude and differences in microphysical growth, especially in the descending phase. Not all models and experiments, however, reach the value of 1: ECHAM5-HAM in Med-19km and Med-18-25km, ULAQ-CCM in Med-19km, and ECHAM6-SALSA, SOCOL-AERv2 and UM-UKCA in all experiments never do. This is due to the use of monthly averages for our analyses and the faster removal, near the tropopause, of sulfate aerosol and $SO_2$ not yet converted to aerosols, especially in Med-19km and Med-18-25km experiments. To confirm this, we observe that this is particularly evident in Med-19km with the lowest injection height. The curves of the experiments with injection between 21-23 km coincide in the build-up phase and the differences emerge later, after 1992: the aerosol lifetime decreases with increasing mass of $SO_2$ injected (table S2), which corresponds to the increase of the aerosol size in all models. In UM-UKCA, the lifetme is increased by one to two months for the meridional-spread emission compared with the point injection. In ECHAM6-SALSA the lifetime increases when increasing the injected $SO_2$ mass. However, Figure 3 and S1 shows that the differences in results between ensemble members of the same scenarios are larger in ECHAM6-SALSA than in other models. This indicates that differences in aerosol lifetimes between Low-22km, Med-22km and High-2km scenarios are probably not statistically significant in ECHAM6-SALSA. Figure s11 panel a shows the sulfate burden from SOCOL-AERv2 for the Low-22km experiment calculated with two model vertical resolutions. This figure further confirms the faster removal of volcanic sulfur during the first months after the eruption in SOCOL-AERv2 even in the 22 km injection experiments. The lower vertical resolution version shows much lower burden peak already in the late 1991, while the higher resolution version peaks at exactly the emitted amount of 5 Tg-S plus the background value of ∼0.17 Tg-S and maintains this peak till early 1992. This is an effect of increased vertical diffusion in the lower resolution version, which quickly redistributes the volcanic cloud vertically in both directions. This brings some of the volcanic sulfur mass closer to the tropopause and the shallow branch of the Brewer-Dobson circulation, reducing its confinement in the tropical reservoir and enhancing removal from the stratosphere (Brodowsky et al., 2021). This agrees with the results of 22km experiments of high-resolution ECHAM5-HAM, which also maintain the emitted amount for some months after the eruption (Fig. 6).

Among all models and experiments, the shortest e-folding time of the global stratospheric sulfate burden is 8 months for EMAC, ranges between 10 and 14 months for ECHAM6-SALSA, ECHAM5-HAM, SOCOL-AERv2 and ULAQ-CCM, and reaches the highest values for UM-UKCA with values between 17 and 23 months, which more closely matches those of HIRS and SAGE-3$\lambda$ of 21 and 20 months, respectively. The e-folding time of the tropical stratospheric sulfate burden is 12 and 13 months in HIRS and SAGE-3$\lambda$ and half for the models with the exception of ECHAM5-HAM for Low-22km, Med-22km and Med-18-25km with a longer duration of 9 months, and UM-UKCA for which it varies between 8 and 14 months, based on the experiments and injection strategy. No model except UKCA can reproduce the observed slow descent phase during 1992 of the stratospheric sulfate burden, and only the High-22km scenario approaches for these models the measured values at the end of 1992, while strongly overshooting them in the preceding months.

Overall, we find that Low-22km and High-22km are the experiments that, in all models, better reproduce the observations in the build-up and descent phase, respectively (fig. 5, s6). The spatial-temporal development of the sulfate burden (Fig. S6) reflects in general that of the AOD (Figs. 2, 3). In the SH, the stratospheric burden shown in SAGE-3$\lambda$ is not reproduced by the models in Low-22km, therefore more $SO_2$ (High-22km) must be injected for the aerosol cloud to persist for as long as in SAGE-3$\lambda$ and reach the same values. This way, however, the burden in the NH is overestimated (Fig. S5). There are clear differences in the position of the stratospheric AOD peak, which lies between 5-20°N in the models, but around 5°S-10°N in the observations pointing to differences in the meridional transport in the early phase after the eruption (Fig. 2). In addition, Figure s11 panels b-c illustrates that the volcanic aerosol mass redistribution between the hemispheres could also be affected by the vertical resolution of the models, because it affects the timings of tropical confinement and across-tropopause removal.

In order to discuss the meridional transport, Figure 7 shows the aerosol mass fraction of the simulated sulfate burden in the tropics (20°N-20°S), in the northern mid-latitudes (35°-60°N) and in the southern mid-latitudes (35°-60°S) with respect to the global value, for SAGE-3$\lambda$ (black line), for all models and scenarios (first row for the different injections amount, second row for the different injection altitudes). Tropical confinement (panels 7a and d) as shown in the observations, is not captured by ECHAM6-SALSA, ECHAM5-HAM, SOCOL-AERv2 and EMAC which underestimate the tropical aerosol mass fraction, resulting in a stronger transport to the NH for the first three models and to the SH for EMAC. ULAQ-CCM overestimates the fraction during the first six months after the eruption and becomes comparable thereafter. UM-UKCA shows tropical confinement comparable to that of SAGE-3$\lambda$ for the 21-23km injection experiments for point injection and shallow and deep injection for merdional-spread emission, otherwise underestimated or overestimated in the other experiments, respectively. However, the similarity between observations and the 21-23km injection experiments for the UM-UKCA point injection masks the lack of aerosols in the southern tropics (0-20°S) and an higher load in the northern extratropics (0-20°N). Indeed, the fraction of burden for the NH midlatitudes (panels 7b and e) is overestimated with differences of up to 20% compared to SAGE-3$\lambda$ (panel 7h) while for the SH (panels 7c and f) it is underestimated but to a smaller extent with differences of 10% compared to SAGE-3$\lambda$ (panel 7i). Same happens for ECHAM6-SALSA, ECHAM5-HAM, SOCOL-AERv2. Overall, NH transport is favoured in all models at the expense of tropical confinement.

In most models, varying the injected $SO_2$ mass does not affect the fraction of aerosols transported out of the tropics towards both hemispheres (panels 7a, b and c). The only exception is ECHAM6-SALSA, where an increased injected $SO_2$ mass increases the tropical confinement, especially in the first six months after the eruption. All models, except ULAQ-CCM, show that the tropical confinement is reduced in favour of transport towards both hemispheres when $SO_2$ is injected below 20 km (Med-19km). Compared to high altitude injection settings (>20 km), Med-19km has the greatest transport in SH. The increase of altitude of injection (Med-22km and Med-18-25km) produces a higher confinement in the tropics with a consequent reduced transport toward both hemispheres in ECHAM6-SALSA, SOCOL-AERv2 and UM-UKCA. In ECHAM5-HAM, the strongest confinement is achieved in Med-22km, while Med-18-25km shows a similar behaviour to Med-19km as most of the sulfate aerosols found below 20 km. In ULAQ-CCM differences among the injection settings emerge six months after the eruption and the injection at lower altitudes (Med-19km) shows a more efficient polewards transport, especially towards the NH.

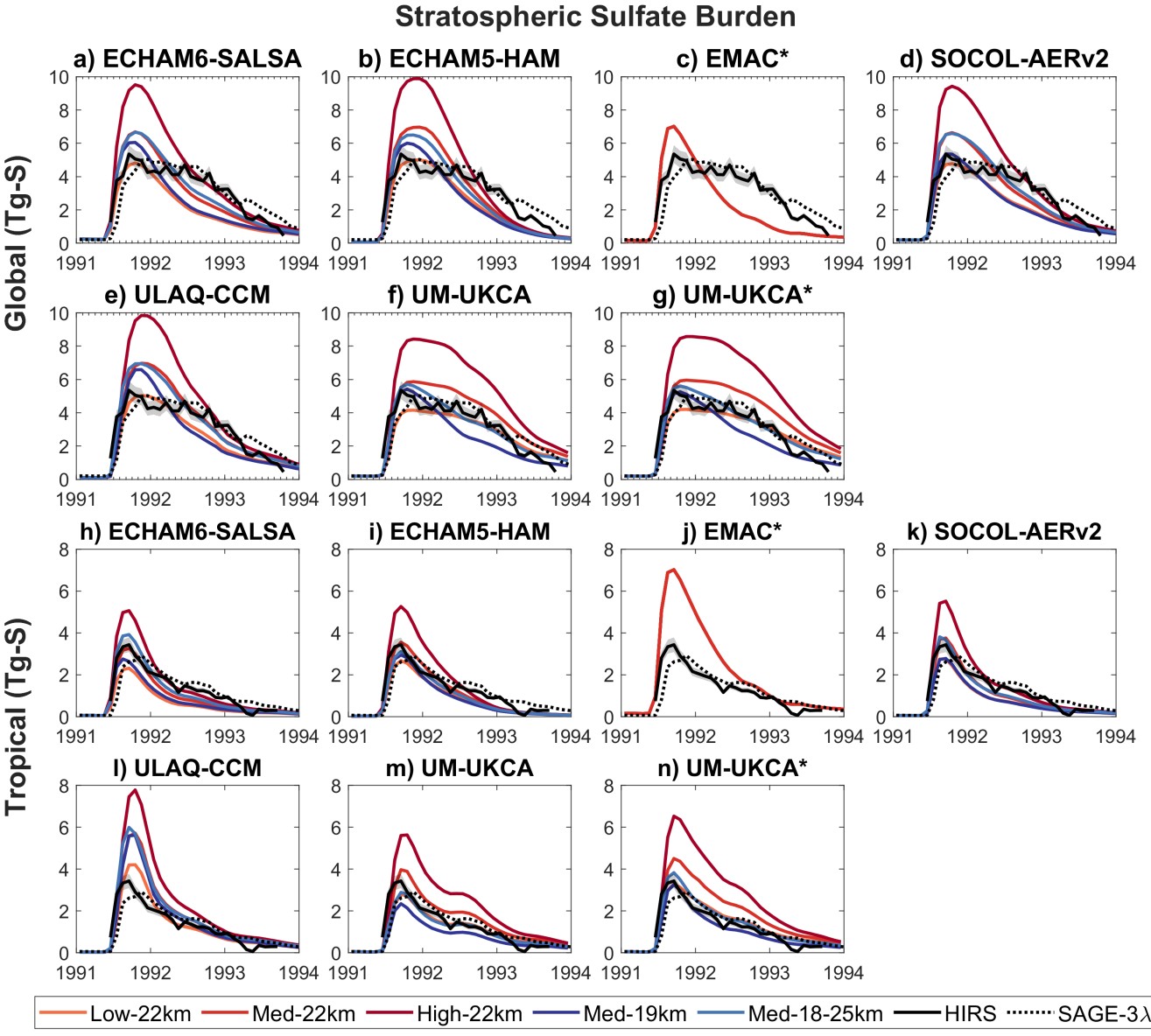

**Figure 5.** Time evolution of monthly values of global and tropical stratospheric sulfate burden in Tg-S (first and second column, respectively). Each panel refers to the respective model in which the different results of the experiments (coloured lines; different line styles for different experiments, see legend on the left) are compared with the HIRS and SAGE-$3\lambda$ data sets (black lines, see legend on the right). * highlight models with spatially spread $SO_2$ injections.

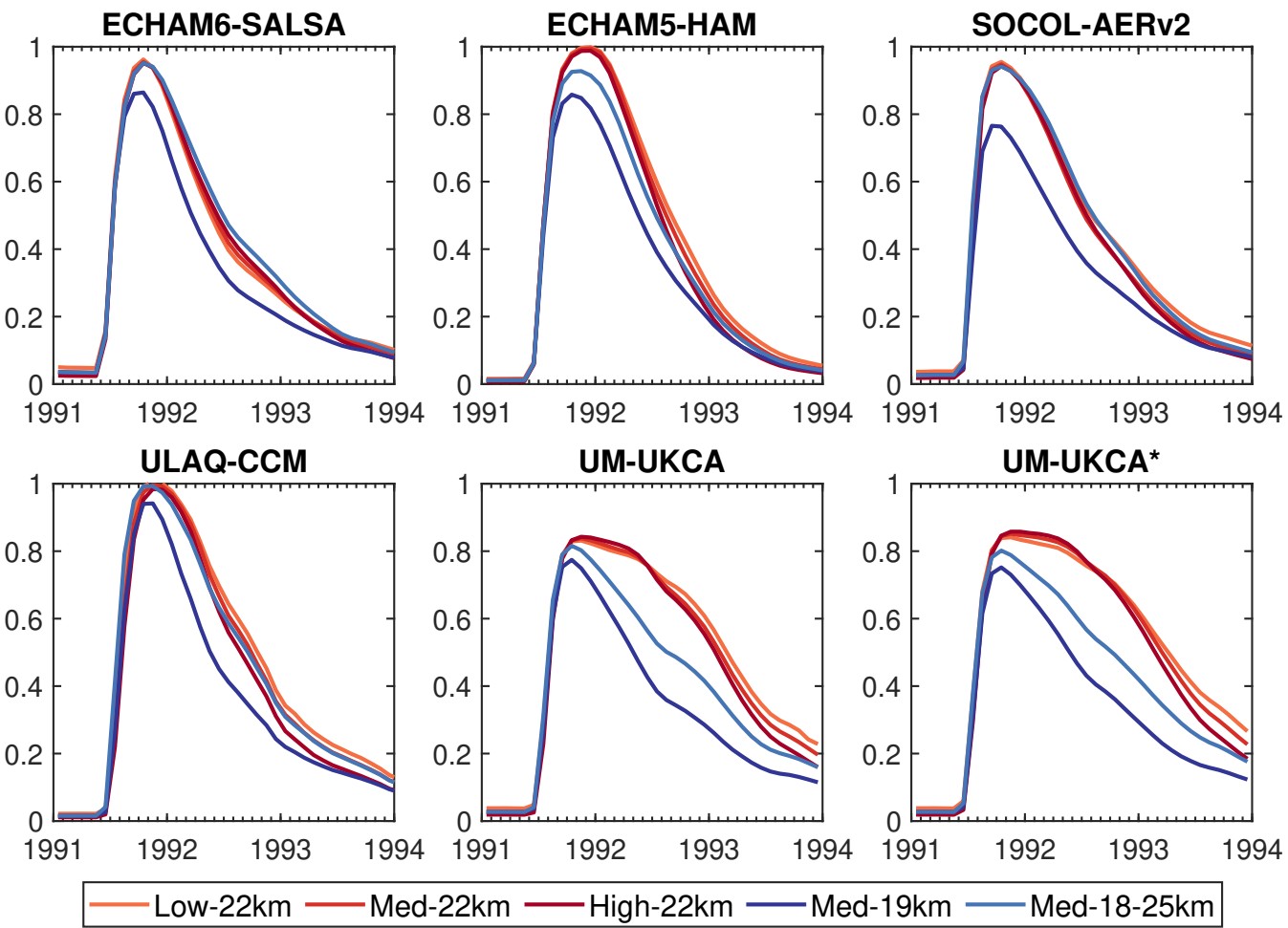

**Figure 6.** Time evolution of global stratospheric sulfate burden normalised to the amount of injected SO$_2$. Each panel refers to the respective model in which the different experiments are compared.

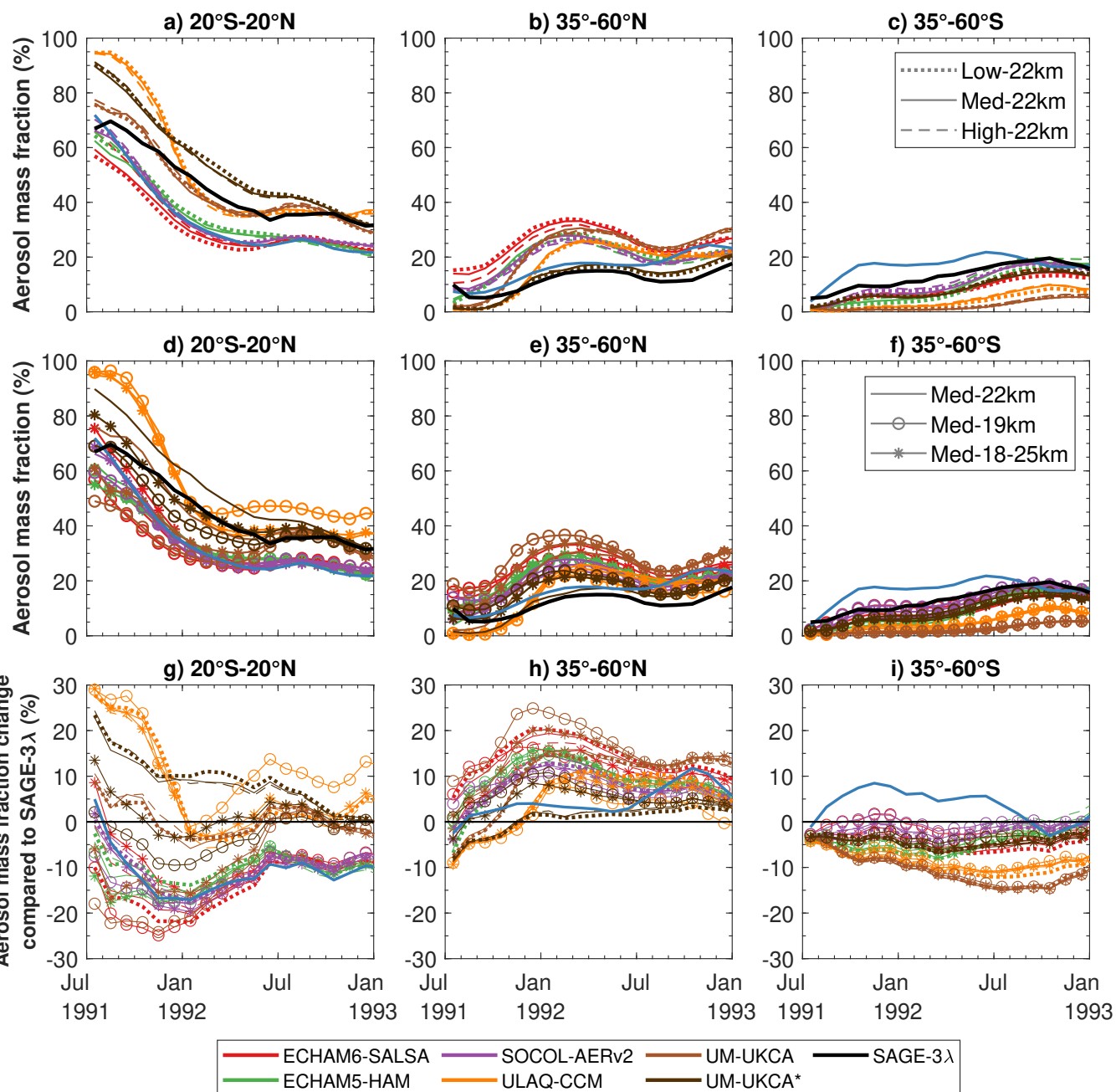

**Figure 7.** Time evolution of the latitudinal partition of the stratospheric sulfate burden. The aerosol mass fraction is calculated with respect to the total burden, for the tropical burden (20°N-20°S, first column, a, d, g), the burden integrated over the northern mid-latitudes (35°-60°N, second column, b, e, h) and over the southern mid-latitudes (35°-60°S, third column, c, f, i). The first row includes the experiments with different amounts of SO$_2$ injected, the second row experiments with different injection altitudes. The third shows the percentage change of the latitudinal partition for all model experiments compared to SAGE-3$\lambda$. Experiments are identified here with different line styles, the different colors refer to the models. * highlight models with spatially spread SO$_2$ injections.

## 3.3 Effective radius and Surface area density

Figure 8 shows the time evolution of the observed and simulated stratospheric effective radius in the tropics (20°S-20°N) and over Laramie (41°N-105° W) (calculation of the effective radius and error bar in Appendix A2). In the tropics (Fig. 8 a-g) the stratospheric effective radius is calculated as the average between 21-27 km because of a paucity of tropical measurements below 21 km in SAGE II. Over Laramie (Fig. 8 h-n), the stratospheric effective radius is defined as the average between 14-30 km in order to compare it with in-situ OPC measurements (Deshler et al., 2019). Models results are calculated as the value of the nearest grid-cell to Laramie; therefore, the ability to reproduce the OPC measurements is more influenced by atmospheric circulation patterns as zonal mean comparisons discussed beforehand and depends also on the horizontal resolution (see Table S1).

Before the eruption, the simulated evolution of the tropical mean effective radius in most models is almost steady compared to SAGE II. Only ULAQ-CCM reproduces the observed seasonal variation and matches the pre-eruption measurements, resulting in particles with a radius of 0.27 $\mu$m, similar to SAGE II (calculated over the 5 months before the eruption). The other models have smaller background particles with a constant value of 0.14 in ECHAM6-SALSA, 0.17 in ECHAM5-HAM, 0.17 in EMAC, 0.15 in SOCOL-AERv2 and 0.10 in UM-UKCA. Over Laramie, ECHAM6-SALSA, ECHAM5-HAM, EMAC and SOCOL-AERv2 have comparable radii to the OPC ones, while ULAQ-CCM and UM-UKCA lay outside the uncertainty range with larger and smaller radii, respectively. The causes of these differences are unclear; however, an in-depth exploration of the background behaviour is out of scope of this paper, and need to be addressed by studies specifically designed to study aerosol microphysics and transport under volcanically quiescent conditions such as the ISA-MIP Background experiment (Timmreck et al., 2018).

After the eruption, all models are able to capture the same decay rate as the SAGE II measurements, remaining flat around the peak reached approximately after October 1991. Most produce a comparable tropical effective radius for about a couple of years, based on different injection settings. The models agree that particle size increases with increasing the injected SO$_2$ mass, with differences from the medium injection scenario within 15% in ECHAM6-SALSA and 10% in ECHAM5-HAM, SOCOL-AERv2, ULAQ-CCM and UM-UKCA. The differences are larger when comparing different injection altitude scenarios and corresponding increase of the particle size is model-dependent. In ECHAM6-SALSA and SOCOL-AERv2, High-22km shows a tropical stratospheric effective radius within 10% of SAGE II until the end of 1993, peaking, respectively at 0.47 and 0.49 $\mu$m compared to 0.51 in SAGE II. In ECHAM5-HAM, all experiments except High-22km, which fits best the observed AOD (see Section 3.1), produce similar effective radii, ranging between 0.46 and 0.51 $\mu$m, and are comparable with SAGE II until the end of 1992. High-22km differs by larger radii reaching a maximum of 0.56 $\mu$m. One year after the eruption, the differences among the different ECHAM5-HAM experiments disappear and the effective radius decreases more rapidly than in SAGE II. EMAC peaks at 0.33 $\mu$m in October and radii stay around 0.30 $\mu$m for less than one year. The low bias hides the faster decrease of the effective radius at about 22 km altitude than in most other models while in the stratosphere below it is similar to observations. In ULAQ-CCM, the effective radius of Med-19km reproduces the SAGE II measurements with a similar time decrease, as differences stay within 10% until the end of 1995, while other experiments produce larger particles, with peaks

ranging between 0.53 and 0.71 $\mu$m. In UM-UKCA, the growth of the effective radius is slower compared to other models, particularly for point injection, but both injection strategies shows the slowest decay which is closest to that of SAGE II. After peaking at different times, the radii between of the two injection strategies are similar and range between the smallest value of 0.10 for Med-19km and the largest value of 0.49 in High-22km, which is comparable with the observations.

Over Laramie, all experiments of ECHAM6-SALSA, SOCOL-AERv2 and UM-UKCA produce radii within the estimated uncertainties of the OPC measurements for all five years in the first two models and after the end of 1991 in UM-UKCA. ECHAM5-HAM and EMAC show comparable values during the pre-eruption phase but in ECHAM5-HAM radii rise faster compared to the observation during the build-up phase while in EMAC, after reaching a peak that is about 30% smaller than that of OPC, the radii assume the smallest values, below the uncertainty. In ULAQ-CCM, all experiments overestimate OPC measurements until early 1992, in particular Med-19km peaks at 0.78 $\mu$m in November 1991, and the effective radius remains at the upper extreme of measurement uncertainty from there on. Increased vertical resolution calculations with SOCOL-AERv2 reveal no difference to the aerosol size before and 1.5 years after the eruption compared to the reference configuration (Fig. s11 panels f-g). During the period of the tropical residence, however, the effective radius noticeably increases due to more aerosol staying in the tropics and the stratosphere and thus available for coagulational growth.

Figure 9 summarises the information regarding the vertical distribution of the effective radius, SAD and extinction at 0.5 $\mu$m for the Med-22km experiment, in the tropical area (20°S-20°N) and over Laramie, six months after the eruption. A corresponding figure including all available experiments is shown in Figure s10. By looking at the vertical profiles of various quantities, biases that are hidden in integrated variables emerge. Figure 9 panel c reveals that the vertical profiles differ not only between models and observations but also strongly between the observations themselves.

In the tropics, the effective radius peaks between 100-50 hPa in ECHAM6-SALSA, EMAC and ULAQ-CCM and between 50-20 hPa in ECHAM5-HAM and UM-UKCA as in SAGE II, with values within 30% of that measured, except for ULAQ-CCM where the radii are up to 4 times larger. In UM-UKCA, the peak of SAD for point injection is centred at higher altitude, around 30 hPa compared to 20 hPa for meridional-spread emission, and with smaller values. SOCOL-AERv2 shows good agreement with SAGE II between 100-20 hPa with values that remain constant around 0.44 $\mu$m above 70 hPa. The tropical SAD simulated by the models follows the same vertical distribution as that of SAGE II, and all models have a peak between 50-20 hPa, with the exception of EMAC whose peak is around 50 hPa. In that range of altitudes, the values of the SAD is comparable with the observations for SOCOL-AERv2 and ULAQ-CCM for most of the attitudes, and is up to 2 times larger in the other models.

The tropical extinction follows the same distribution of the SAD. In this case, the extinction is compared with SAGE II and GloSSAC and large differences exist between them: below 20 hPa the extinction in GloSSAC is larger than in SAGE II and the differences increase with decreasing height up to 100% compared to SAGE II because of its gap-filling with ground-based measurements (Thomason et al., 2018; Kovilakam et al., 2020). Above 70 hPa, around the lower bound of the injection altitude, models extinction is even larger than GloSSAC: ECHAM6-SALSA, SOCOL-AERv2 and ULAQ-CCM approaches the measurements at limit of maximum of uncertainty around 70-25 hPa, EMAC between 40-20 hPa, while ECHAM5-HAM and UM-UKCA overestimate measurements up to twice their value. Below 70 hPa, all models underestimate the GloSSAC data,

but the models extinction is still larger than that of SAGE II, with the exception of EMAC, which shows the greatest extinction below 50 hPa, where it peaks. Considering that the SAD depends on the size and the number of particles, we can assume, for the models that show a comparable radius and a larger SAD compared to SAGE II in the tropics, that they overestimate the number of optically active particles and therefore show a larger extinction (ECHAM5-HAM and UM-UKCA).

Over Laramie, the vertical distribution of the effective radius is within the error bar of the OPC measurements up to 20 hPa in ECHAM6-SALSA, ECHAM5-HAM, and SOCOL-AERv2, while ULAQ-CCM produces larger particles especially below 50 hPa. In EMAC the effective radius is at the lower limit of the uncertainty but is the only model able to reproduce the vertical profile of the SAD from OPC measurements in terms of the position of the maximum and values. The models that showed faster transport in the northern mid-latitudes overestimate the observed SAD for most of the altitudes.

The ability to reproduce the observations also depends on the period considered (fig. s8 and s9): in the first months after the eruption models and observations show large differences, especially for SAD and extinction, which are overestimated at both latitudes considered. This may be related both to the sensitivity to the actual meteorological conditions that climate models are unable to accurately replicate, and to the absence in HErSEA simulations of volcanic ash injection that could remove some of the initial $SO_2$ gas or affect the local winds and the $SO_2$ dispersion (Ayris et al., 2013; Zhu et al., 2020; Dhomse et al., 2020; Kloss et al., 2021; Niemeier et al., 2021). This sensitivity to the initial conditions of $SO_2$ injections decreases the more time passes after the eruption. One year after the eruption, the models still show a vertical profile of the effective radius comparable to observations, while the simulated SAD starts to decrease everywhere after six months from the eruption, underestimating tropical values but still overestimating OPC measurements.

## 4 Discussion

With the use of Taylor diagrams, we highlighted the experiments that better match the observations in terms of stratospheric AOD, in two different time periods, based on the reliability of the measurements. Each model requires different injection scenarios to reproduce the observations, due to differences in the transport and microphysical processes and their mutual interaction. Even considering the best set of initial parameters based on AOD (Fig. 2), differences with observations more or less persist in the models, and we can not unequivocally define a "best" model as that varies depending on the variable considered and the timing of the observation.

Comparing the results of the models between the experiments with the same injection setup, we observe a large difference between models in reproducing the stratospheric optical depth compared to the similar evolution of the global stratospheric sulfate burden. It is hard to disentangle the transport and the microphysics contribution on the differences in the considered variables, i.e. what fraction of it depends on microphysical schemes or different dispersion of the aerosol cloud. We first considered the contribution of $SO_2$ oxidation by OH to differences in the timing of the peak for the stratospheric sulfate burden (Fig. 5) and, consequently, AOD (Fig. S2). For models with prescribed OH, differences in the stratospheric rate of $SO_2$ conversion may depend on the injection altitude, due to an earlier removal through the tropopause flux when the injection is closer to the tropopause. For models with interactive OH we observe a longer e-folding time for higher mass of $SO_2$ injected and

## Stratospheric Effective Radius ($\mu$m)

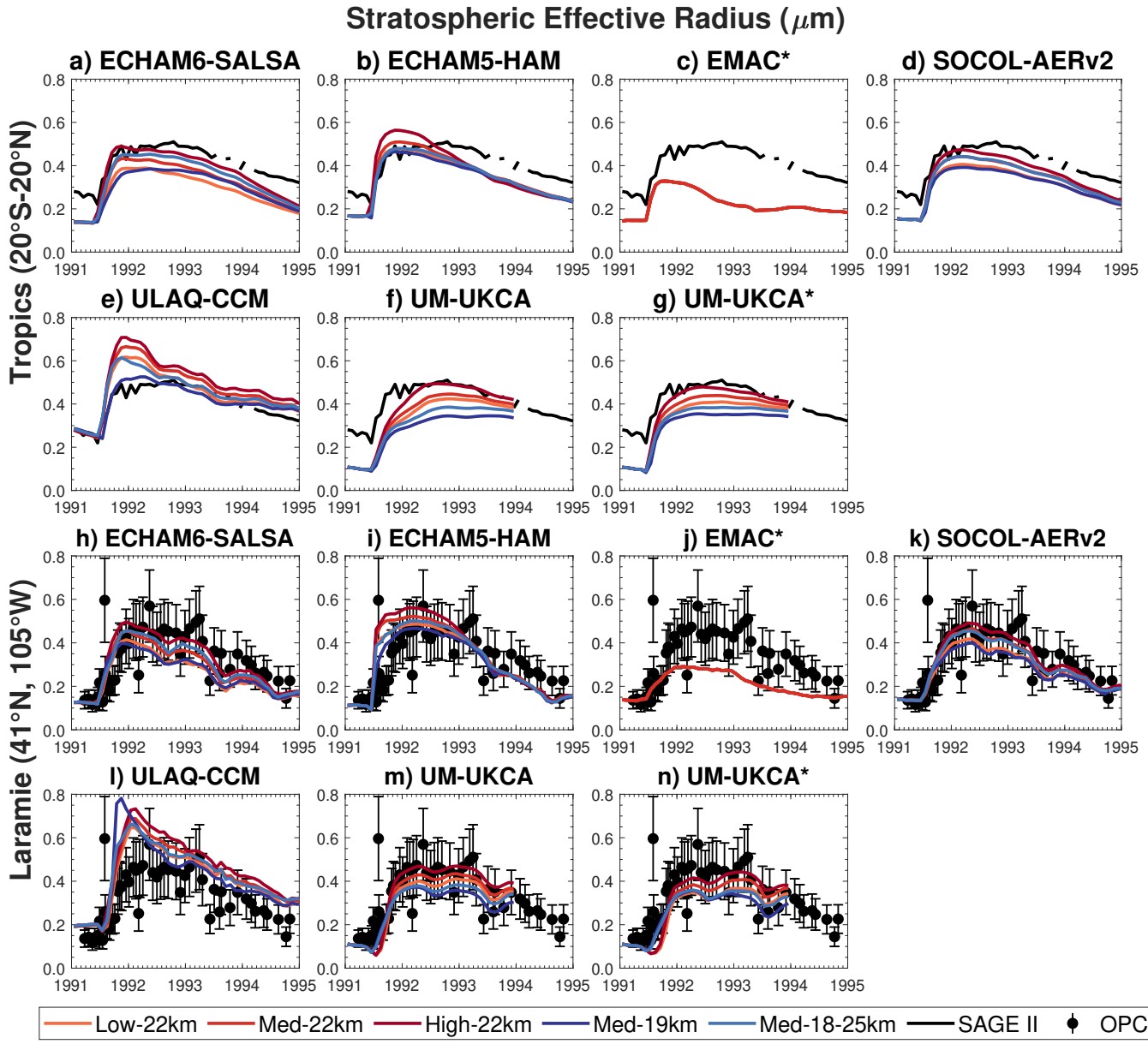

**Figure 8.** Time evolution of stratospheric effective radius ($\mu$m) in the tropics (panels a-g) and over Laramie (41°N, 105° W, panels h-n). In the panels of the first row, the stratospheric effective radius of the models is calculated between 21-27 km (50-20 hPa) to be compared with the available SAGE II observations. In the panels of the second row, it is calculated between 14-30 km (130-10 hPa) to be compared with the OPC observations. * highlight models with spatially spread $SO_2$ injections.

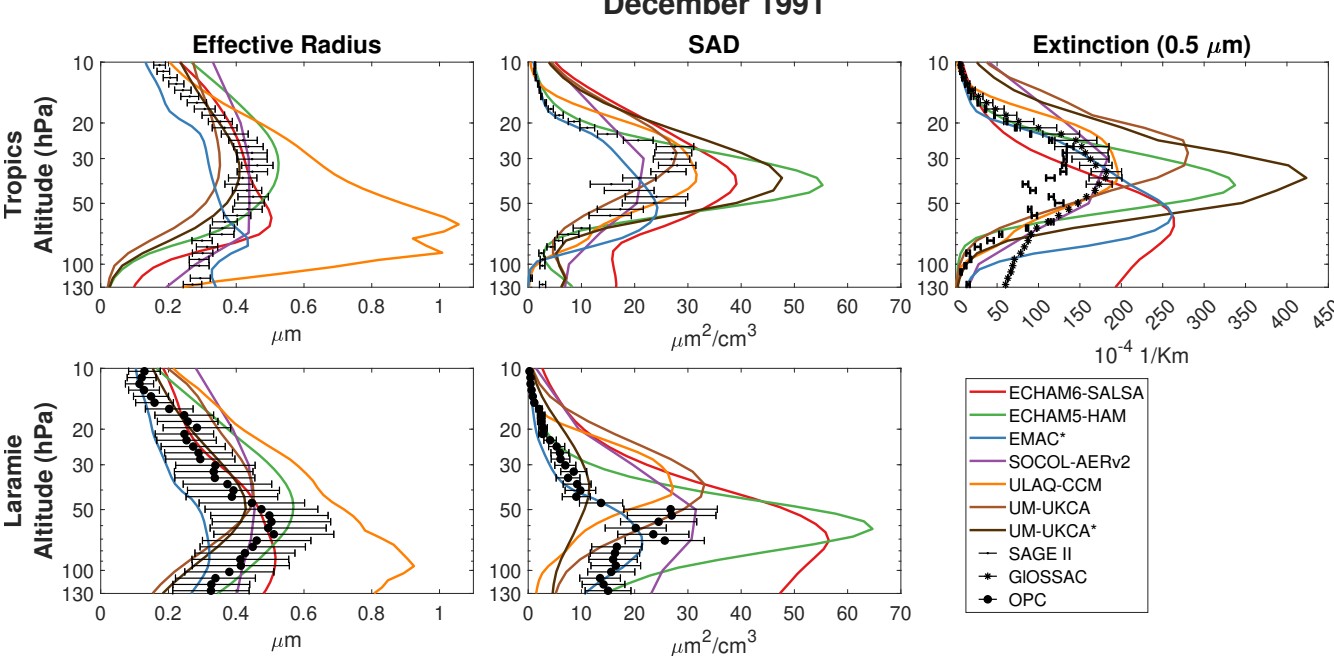

**Figure 9.** Vertical profile of the effective radius in $\mu m$ (left panels), surface area density (SAD) in $\mu m^2/cm^3$ (middle panels), and extinction at 0.5 $\mu m$ in 1/km (right panel) in the tropics (first row) and over Laramie (second row) for Med-22km in December 1991. Model results are compared with SAGE II and GloSSAC in the tropics and with OPC over Laramie. * highlight models with spatially spread $SO_2$ injections.

when injected in a narrow altitude range (Med-22km vs Med-18-25km). Due to the availability of only monthly values, some observations of the $SO_2$ behaviour at a finer resolved temporal scale are not possible here. Furthermore, since the lifetime of sulfate depends on both OH concentration and transport and mixing into adjacent grid boxes, when comparing different

models, the timing of the peak cannot be simply related to the treatment of OH.

However, we find a common problem in transport, either too fast from the tropics to high northern latitudes (ECHAM6-SALSA, ECHAM5-HAM, SOCOL-AERv2), confined in the NH (UM-UKCA for point injection), or too confined to the tropics (ULAQ-CCM). The different tropical confinement can be affected by a different vertical advection scheme between ULAQ-CCM and the other models, based on the same dynamical core ECHAM5 or ECHAM6. Here, the tropical confinement

depends on the different horizontal resolution (Niemeier et al., 2020) while the particular definition of the tropical pipe (see Waugh et al., 2018) may also strongly affect this conclusion. The vertical resolution of a model can also affect the transport from the tropics to high northern latitudes: Brodowsky et al. (2021) showed for the SOCOL-AER model that a longer tropical confinement was found with increased vertical resolution. Hence, the transport to NH and SH can depend on model version and injection setting: previous MAECHAM5-HAM simulation of the Mt. Pinatubo eruption Niemeier et al. (2009) show a

similar pattern for the stratospheric AOD compared to AVHRR and SAGE II, by injecting a mid-range amount of $SO_2$ between the Med-22km and High-22km experiments (8.5 Tg-S) into one grid-box at the location of Pinatubo and a model layer around

24 km, but assuming less vertical levels without internally generated QBO. The Typhoon Yunya, which cannot be reproduced with coarse resolution in models, might have played a role in the equatorward transport of the volcanic cloud as well, causing a stronger transport into the SH than in most model results. Better transport to the SH showed EMAC, which has been nudged to the real meteorological conditions and the UM-UKCA version with emissions between 15°N and the Equator.

The meridional transport in the models depends on the vertical wind structure and on the vertical distribution of the simulated volcanic cloud in the first months after the eruption. Labitzke and McCormick (1992), based on SAGE II measurements, showed for the early post Pinatubo period an upper transport regime (above 20 km) in which aerosols remain confined to the tropical reservoir spreading between 30°N and 10°S, and lower transport regime (below 20 km) in which aerosols mainly spread to northern high latitudes. Between August and September, aerosols above 20 km spread across most of the SH, reaching latitudes of 50°S, followed in November and December by an enhancement in the NH due to the transition from boreal summer to winter circulation in the middle and upper stratosphere. Most of the models show a faster transport in the NH is favoured when aerosols are mainly distributed in the lower transport regime (Timmreck et al., 1999). The lower stratospheric part of the injection profiles is also strongly affected by the inconsistencies between the modeled and real tropopause heights at the time of eruption (Brodowsky et al., 2021). This effect can be additionally enhanced in the models with low vertical resolution (Fig. S11). We note that the strength of the meridional transport is also seasonally dependent, and therefore eruptions happening in other seasons would result in different distributions of the aerosol cloud (Visioni et al., 2019; Toohey et al., 2011). We find that the injection rate does not affect the fraction of aerosols transported out of the tropics towards both hemispheres with the exception of ECHAM6-SALSA where an increased injected $SO_2$ mass increases the tropical confinement, especially in the first six months after the eruption. This is probably due to a stronger radiative interaction from the absorption of more long-wave radiation by larger particles. The behaviour of the other models is consistent with the findings of Young et al. (1994) and Aquila et al. (2012) where the aerosol heating by absorption of the infrared radiation induces a lofting and a divergent motion that affects only the initial transport (within one month) of the aerosols towards and within both north and south tropics.

Even when models and measurements look comparable for the integrated variables (Figures 8 and S2), these similarities hide the models inability to reproduce the observed vertical structure depending on the latitude and time period after the eruption under consideration (Figures 9, S8 and S9). Most models take up to six months before they can reproduce the vertical structure of effective radius, SAD and extinction in the tropics, and up to a year at mid-latitudes. The vertical distribution of SAD and effective radius in three moments identifying the build-up, maximum and descent phase of the evolution of the sulfate burden (September and December 1991 and June 1992, respectively) show an initial overestimation of the observations and an underestimation one year after the eruption. The lack of ash co-emission, a process not included in HErSEA simulations, could be crucial in the first days/month to better reproduce the initial cloud evolution (Stenchikov et al., 2021). On one hand, the ash may have removed parts of the initial sulfur cloud through the $SO_2$ or $H_2SO_4$ uptake on these coarse particles, which have a significant fall velocity (Zhu et al., 2020); on the other hand, the presence of smaller ash particles causes greater heating and vertical lofting of the volcanic cloud (Niemeier et al., 2021; Kloss et al., 2021), which could result in slower meridional transport and longer lifetimes of stratospheric volcanic aerosols, depending on the latitude and injection altitude of $SO_2$ (Niemeier et al., 2009; Stenchikov et al., 2021). Aberystwyth lidar measurements from Vaughan et al. (1994) show

a signature of depolarising particles around 16 km between November and December 1991. That corresponds to the sudden enhancement of the SAD from the Laramie measurements and has been identified as ash-rich particles (Pueschel et al., 1994). The faster transport to the northern mid-latitudes in the models than observed may have removed most of the stratospheric particles, so that the aerosol lifetime in the models is about half that observed.

In addition to different transport and microphysical mechanisms, the neglection of the Cerro Hudson eruption in August 1991 that injected about 0.75-2.0 Tg-S of $SO_2$ between 12 and 18 km (e.g. Saxena et al., 1995; Bluth et al., 1997; Neely III and Schmidt, 2016; Carn, 2022) in the simulations, may partially explain the lack of the observed sulfate aerosol in the southern extratropics that we find in all model scenarios. The only exception is EMAC, which included the eruption of Cerro Hudson and nudged the meteorological variables. The importance of the Cerro Hudson eruption has therefore been evaluated with ULAQ-CCM performing two additional simulations that consider the lower and upper estimates of the $SO_2$ injection in addition to the Med-22km experiment. Significant deviations from the results of Med-22km emerge only when including the Cerro Hudson eruption with the injection of 4 Tg $SO_2$ at 12-18km altitudes (Fig. S7 panels c, g, k-n). We observe an increase in the stratospheric sulfate burden and optical depth in the SH that better reproduces the observations for the 2 months following the Cerro Hudson eruptions. However, the shorter e-folding time of stratospheric aerosol for the extra-tropical eruption does not affect the global stratospheric lifetime and is still not sufficient to explain the lack of aerosol in the SH in the following months, which we therefore attribute to transport.

The inter-models differences may depend on numerous factors that interact with one another; this makes it hard to group models by perceived similarities, for instance a similar modal scheme, similarities in the large scale transport or an absence of interactive stratospheric chemistry. Laakso et al. (2022), for instance, used the same climate model (ECHAM-HAMMOZ) with two different aerosol microphysics schemes, one sectional and one modal. Even just this difference produced an effective radius up to 52% greater in the sectional scheme than in the modal scheme simulation for the same amount of injected $SO_2$. Further, Niemeier et al. (2020) showed that, in two models with a similar modal scheme but different vertical advection (CESM-WACCM-110L and MAECHAM-HAM), the resulting vertical distribution of the aerosol cloud can be substantially different. Even in the same model (CESM1-WACCM), Richter et al. (2017) showed that the presence or not of interactive chemistry could strongly affect the local stratospheric warming, and thus the residual vertical velocity changes, due to feedback from the changing ozone. In our case, all of these differences are compounded, therefore it is hard to identify which exactly is the cause of the disagreement. Furthermore, in all the works cited above, $SO_2$ was injected continuously for a number of years rather than in an impulsive way, whereas in the case of a volcanic eruption, the synoptical conditions at the time of the eruption play an important role (Thomas et al., 2009; Toohey et al., 2014; Niemeier et al., 2021; Jones et al., 2016). In our case, the experimental protocol requires the consistency of the QBO with observations through the post-eruption period; nonetheless, there are smaller scale processes and variability that are not reproducible by models with a coarse resolution that would affect the initial state of the system, as the formation of mesocyclone during the first day after the eruption (Chakraborty et al., 2009) or the passage of Typhoon Yunya within 75 km northeast the eruption (Oswalt et al., 1996).

## 5 Conclusions

The ISA-MIP HErSEA experiment protocol was designed to investigate the differences and the consensus among a group of climate models, all with interactive stratospheric aerosol microphysics, by comparing them with measurements after the Mt. Pinatubo eruption in 1991. This is done through a well-defined experimental protocol with different sets of initial parameters for the stratospheric $SO_2$, both in terms of magnitude (5, 7 or 10 Tg-S injected) and altitude of the $SO_2$ cloud (18-20, 21-23, 18-25 km, uniformly distributed). One important finding from this intercomparison is that there is now a general consensus among the models that an $SO_2$ emission amount at or below the lower-end of the observed stratospheric $SO_2$ mass-loading (14-23 Tg) are required to reproduce the observed sulfate aerosol loading from that time period. However, the set of injection parameters that best fits the observation changes in some models depending on the variables to be considered (aerosol optical depth, effective radius, sulfate burden, surface area density).

The main reason for the disagreement with observations is stratospheric transport, which is too fast towards the northern mid-latitudes for some models or results in stronger tropical confinement in others. The transport consequently influences the growth of sulfate aerosols and their global distribution, which in turn affects the persistence of aerosols in the stratosphere, with a feedback on the transport itself (Brühl et al., 2015; Niemeier and Schmidt, 2017; Visioni et al., 2018b). Other reasons could be related to the absence of processes such as the absence of the Cerro Hudson eruption in the southern extratropics two months after the Pinatubo eruption, which may partly explain the initial lack of sulfate aerosols in the southern hemisphere, and the omission of ash injection which would be crucial in the early days/months to better reproduce the initial evolution of the cloud. Our results highlight the need for some specific experiments that might be needed to disentangle the different components that contribute to the overall uncertainty. For instance, simulations that nudge stratospheric transport to reanalyses (as done in Schmidt et al., 2018, in CESM(WACCM)) in multiple models could clarify the role of different microphysical schemes. Similarly, consistently turning interactive stratospheric chemistry on and off in multiple models could highlight the importance of ozone feedback (as done in Richter et al., 2017). Last but not least, dedicated tracer experiments of an idealized volcanic cloud excluding chemical microphysical processes are necessary to asses the role of stratospheric transport in the models.

Overall considering the best set of initial parameters, differences between models and observations remain and the inter-model differences are still large, as found before in other multi model-experiments of explosive volcanic clouds (i.e. Tambora in Marshall et al., 2018; Clyne et al., 2021). We also note that the observations themselves show disagreement, sometimes as high as inter-model differences, because of various issues with the saturation or sensitivity of the particular instrument. Our observations around the reliability of the measurements during the Pinatubo event highlights the future need for more observation in order to be better prepared for future explosive volcanic eruptions (Newhall et al., 2018; Marshall et al., 2022), both for understanding short and long term impacts and as a benchmark test for current Earth System models. In the absence of large volcanic eruptions in the early 21 century, where a wealth of observational data exist it might therefore be also an alternative to focus in future aerosol model intercomparisons studies on moderate eruptions, e.g. the Raikoke eruption in 2019.

As a first study of the inter-model differences within ISA-MIP HErSEA, we focused on the aerosol optical depth and the variables on which it depends, such as the loading and size of the sulfate aerosols. Therefore, we suggest for follow-up studies the comparison of radiative forcing and ozone changes, which immediately follow the analyses done here.

*Data availability.* Simulation data are available at https://doi.org/10.7298/MM1S-AE98.

## A    Analysis of model output

### A1    Taylor Diagrams

In section 3.1 we use Taylor diagrams (Taylor, 2001) in order to summarise all the information regarding the reproducibility of the stratospheric optical depth simulated compared to satellite observations. Taylor diagrams provide a concise statistical summary of how well patterns from simulations and observations match each other in terms of their correlation (COR, azimuthal angle), their root-mean-square difference (RMSD, proportional to the distance between the observations - grey and black circles on the x axis- and experiments - colored circle), and the ratio of their variances (STD, x and y axis). STDs, RMSs and CORs are calculated for zonal values of the stratospheric AOD for two different time periods (first year and second year after the eruption). Therefore, similar STD, higher COR and lower RMSD mean similar amplitudes of variation in terms of latitudinal distribution and time evolution.

### A2    Effective radius

The effective radius is calculated as the ratio of the third and second moments of the number size distribution of the aerosol particles. This results in Eq. A1 for models with a sectional scheme; in this case, the sum is over the bins and $n_i$ is the number of particles and $r_i$ is the radius of particles in each bin. In models with a modal scheme, the effective radius is calculated as the sum over the modes as in Eq. A2, where $SAD_i$ is the surface area density and $vol_i$ is the volume density. In EMAC (modal scheme) the quantity is estimated from the median radia for accumulation and coarse mode particles since it was not stored in the output.

$$reff = \frac{\sum_i n_i \cdot r_i^3}{\sum_i n_i \cdot r_i^2} \tag{A1}$$

$$reff = \frac{3 \cdot \sum_i vol i}{\sum_i SAD i} \tag{A2}$$

The stratospheric effective radius (reff$_{strat}$) for the models and SAGE II is calculated in Eq. A3 by integrating the provided effective radius (reff) from the tropopause to the top of the atmosphere weighted with the SAD. The thickness of the vertical layer (h) is calculated from the hypsometric equation (Eq A4)

$$reff_{strat} = \frac{\sum_z (SAD \cdot h \cdot reff)_z}{\sum_z (SAD \cdot h)_z} \tag{A3}$$

$$h = \frac{R \cdot T}{g} \cdot \ln \frac{P_{z+1}}{P_z} \tag{A4}$$

For the OPC measurements, we calculate the stratospheric effective radius (Eq. A5) as in Kleinschmitt et al. (2017) for the updated UWv2.0 data set. The measurement error bars consider a 40% uncertainty in SAD and vol and assume a correlation coefficient of 0.5 between SAD at different altitudes, vol at different altitudes and SAD and vol at the same altitude.

$$reff_{strat} = \frac{3 \cdot \sum_z vol_z}{\sum_z SAD_z} \tag{A5}$$

*Author contributions.* IQ led the analysis and wrote the paper with contributions by CT, UN and DV. CT, GM, CBru, and SD designed the study. HF ran the ECHAM5-HAM simulations and provided the output data. TS, ER, and CBro ran the SOCOL-AERv2 experiments and provided the output data. All authors contributed to discussion and finalisation of the article.

*Competing interests.* The contact author has declared that neither they nor their co-authors have any competing interests.

*Acknowledgements.* Claudia Timmreck and Ulrike Niemeier were supported by the Deutsche Forschungsgemeinschaft Research Unit Vol-
775 lImpact (FOR2820 (grant no. 398006378)) and use resources of the Deutsches Klimarechenzentrum (DKRZ) granted by its Scientific Steering Committee (WLA) under project ID bm855 "ISA-MIP". Timofei Sukhodolov and Eugene Rozanov acknowledge the support from the Swiss National Science Foundation (SNSF) project POLE (grant no. 200020-182239) and the Ministry of Science and Higher Education of the Russian Federation (grant no. 075-15-2021-583). Calculations with the SOCOL-AERv2 model were supported by a grant from the Swiss National Supercomputing Centre (CSCS) under the project S-1029 (ID 249) and by the ETH Zürich cluster EULER. We thank Jennifer
Schallock for providing parts of her PhD-thesis work with EMAC for further analysis in HErSEA.

The paper is part of the "Interactive Model Intercomparison Project" from the WCRP/SPARC activity "Stratospheric Sulfur and its Role in Climate (SSiRC) ".

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
