# Peer review of "Interactive Stratospheric Aerosol models response to different amount and altitude of $SO_2$ injections during the 1991 Pinatubo eruption"

_Atmospheric Chemistry and Physics, 2022_

## Author Comment (AC1)

**Reviewers' comments are in bold.** Authors' responses are in blue.

**Moderate comments**

**MC1) The role of the Cerro Hudson eruption is really an important question. Checking out the latest version of the MSVOLSO2L4 inventory (curated by the NASA and Simon Carn, https://disc.gsfc.nasa.gov/datasets/MSVOLSO2L4_4/summary), the Hudson eruptions injected 4Tg SO2 at 12-18km altitudes. The Neely and Schmidt (2016) inventory reports 1.5Tg SO2 between 11 and 16km. So that would be between 7-40% of the Pinatubo mass depending on which value you consider for Cerro Hudson and for Pinatubo, a big number in any case. Could you repeat simulations, for at least one model and one of your scenarios, with Cerro Hudson included? I UM-UKCAwould actually suggest running one with the lower-end emission (Neely and Schmidt) and one with the upper end emission (MSVOLSO2L4). If you run only one set of parameters for Hudson, I strongly suggest picking a SO2 mass in between these two estimates and not just the lower estimate (which is the one mentioned in your manuscript). Doing this test would really add a lot to the paper.**

We thank the reviewer for his suggestion; however we have to state clearly here that the purpose of the HErSEA Pinatubo experiments was a simulation of the Mt. Pinatubo eruption in June 1991, but not one of the stratospheric aerosol load in 1991. Consequently a model data comparison for the Southern Hemisphere (SH) could potentially be biased due to the simultaneous occurrence of the Chilean Cerro Hudson eruption in August 1991. To test this hypothesis we followed the reviewer's suggestion and performed the two suggested simulations with the less expensive model, ULAQ-CCM, described in section 2.1.1.1. We added a summary figure for the stratospheric sulfate and SO2 burden and the optical depth in the supplementary material (S4) and the analysis of the results in the discussion section. Significant deviations from results of Med-22km emerge only when including the Cerro Hudson eruption with the injection of 4Tg SO2 at 12-18 km altitudes (Figure S7 panels c, g, k-n). While we do observe an increase in the stratospheric sulfate burden and optical depth in the SH that better reproduces the observations for the 2 months following the Cerro Hudson eruption, the shorter e-folding time of stratospheric aerosol for the extratropical eruption does not affect the global stratospheric lifetime and is still not sufficient to explain the lack of stratospheric aerosol in the SH in the following months, which we therefore attribute to transport.

**MC2) The injection strategy in UM-UKCA is really different. My personal experience with this model is that it's hard to get any SH transport unless injection is spread between 0 and 15N as done in your paper, and I think this is documented in published papers by Dhomse, Mann and co-authors. I recommend that you acknowledge this more explicitly in section 2, and that this model is singled out in a similar way to EMAC on all figures (e.g. on Figure 2 add a * symbol like you did for EMAC, and same everywhere else). It would be valuable to add comparison of point vs 0-15N injection for this model, either by running a point injection for one of your scenarios or by using already existing/published runs. In table 1, I would replace "band" for the injection region by "0N-15N, 120E". I believe that "band" injection would be understood by most people in this community as a zonal injection at the volcano latitude following the terminology used in e.g. Zanchettin et al. (2016) and Clyne et al. (2021), so "band" is misleading here.**

We appreciate the reviewer raising this important issue about the injection strategy. Unfortunately, the ISA-MIP/HErSEA protocol only specifies different vertical distributions of the sulfur emission but not the horizontal one. This is a clear weakness of the protocol and complicates a direct comparison of the different model simulations as the initial horizontal distribution of the volcanic cloud has an important impact on the spatio-temporal distribution of the cloud e.g Clyne et al. (2021).

Many of the applied models here injected the sulfur emission only in the corresponding grid box where Mt. Pinatubo is located but some models (EMAC and UM-UKCA) prefer due to various good reasons a different injection strategy. We clarify this now in the revised paper, where we also replace the word zonal by "meridional-spread injection". We also label the different model simulations which have no point injection with some asteriks.

Since the publication of the ACPD version of our manuscript, we have re-run the Dhomse et al. (2020) simulations, from the same start-dumps, which we had stored on the elastic tape system at the UK's JASMIN data storage and compute resource facilities (http://jasmin.ac.uk/about). This enabled us to run the additional HErSEA-Pinatubo shallow-low and deep emission sensitivity integrations that we had not been able to carry out at that particular time. As suggested by the reviewer, we have also carried out a 2$^{nd}$ HErSEA ensemble of integrations, emitting only in the 13.75 to 15.0N gridbox (i.e. just 1 "injection height-range column"). We will discuss the differences between both experimental set up now in the discussion part of the revised manuscript.

We openly confirm that the reviewer is correct, that, without making the adjustment for the initial southward shift of the Pinatubo plume, the UM-UKCA model then does not capture the southward Southern Hemisphere transport with the "free-running approximate QBO" approach. A feature the UM-UKCA model shares with other global aerosol models. Discussion at SSiRC workshops and other international conferences indicate that for initial simulations global aerosol models often need to include a meridional spread to better match satellite observations. The reasons for this are still open and one of the biggest challenges of our community.

**MC3) I see no comment on data availability which is crucial before publication. In particular, having SI tables or a netcdf archive with the processed data displayed on key figures (for both model and observations) would be really welcome (at least for figures 2, 5, 8). This would facilitate comparison to your results for future studies.**

We apologize for the lack of link - we planned to include it during the revision process as the process is not straightforward. All data is now available online and the DOI has been provided in the section "Data availability".

**MC4) This one is more a remark than a comment. This is a really nice paper and I strongly recommend prompt publication, but it's too bad that there aren't more modelling groups that ran the ISA-MIP HErSEA experiment in time for this paper or didn't follow the protocol. Out of the four models that followed the experimental protocol, three have some version of the ECHAM model at their core which limits model diversity especially when bias in circulation and subtropical barrier are suggested to be one of the main challenges. Figure 1 in Clyne et al. (2021) also suggest that the model used in this paper will produce middle-range SAOD estimates.**

**Is there no chance to include results from the IPSL or WACCM groups in this paper? Or to run the UM-UKCA simulations following the experimental set-up of figure 1 including point injection (or at least repeat the med scenario with the same 0-15N injection but a 19km height)? I realize that this is likely challenging at this stage especially for my first question. If so, my only recommendations are to acknowledge a bit more explicitly the lack of model diversity wrt the two points above (ECHAM as core model and middle-range SAOD estimates), and maybe to add a few sentences towards the end of the paper reflecting on what we can do as a community to encourage stronger participation to such MIPs? This could help the community leverage more funding and/or computing resources to support such intercomparison exercises.**

The reviewer is right in pointing this out: we strongly agree that the addition of more modeling groups would be beneficial for these kind of studies. However, given the voluntary nature of model intercomparisons (and the constraints on manpower and computer time, which the reviewer already mentions in his comments!) this was not possible. At the beginning of our study we hoped to include WACCAM results as well but due to the aforementioned reasons it was not possible. We will point to a certain lack of model diversity and include some recommendations in the conclusions related to future participation.

*Note: I realize that running additional simulations as suggested in MC1, MC2 and MC4 requires time and resources. However, the simulations suggested would use the same set-up as the ones already ran for the paper, so I hope that at least some of them are feasible within a reasonable timeframe given the atmosphere-only setup and small ensemble sizes/duration. The order of my comments reflects the priority I'd give to these additional simulations.*

**Minor and editorial comments**

**Line 3: Replace "plume" by "cloud" (here and throughout the paper). You mostly use "cloud" later, and "plume" is very commonly used for the vertically rising column rather than the large-scale horizontally (mostly) spreading cloud.**

 Corrected.

**Line 17: The link with ash will not be obvious to a non-expert reader, could you contextualize briefly?**

We changed the sentence to

"This draws attention to the importance of including processes such as the ash injection for the removal of the initial $SO_2$ and aerosol lofting through local heating."

**Line 18: add the country or latitude in parenthesis after "Cerro Hudson" so that the link is easier to make for non-expert readers.**

We have removed the reference to Cerro Hudson here due to the changes made to the paper with the new results of the additional simulations carried out.

**Line 22: delete "can"** Corrected.

**Line 29: "framework" instead of "frame"?** Corrected.

**Line 30 and section 1: you have many paragraphs that are 3-5 line long; consider grouping some of them.** Corrected.

**Lines 36-38: you could maybe point to earlier measurements and more recent papers to contextualize both the SO2 and ash injection height. Fero et al. (2009, https://doi.org/10.1016/j.jvolgeores.2009.03.011) seems particularly relevant. The IVESPA database (http://ivespa.co.uk/, endorsed by IAVCEI) also has best estimate and uncertainties based on extensive literature compilation for many events including Pinatubo. For Pinatubo the height of the plume top, ash injection height and SO2 injection height are 32+/-3 km asl, 22+/-3 km asl and 25+/-3 km asl.**

We thank the reviewer for his suggestion. We have now included the reference to the suggested paper and extended the corresponding paragraph in the introduction:.

*"Mount Pinatubo is located in the western part of the island of Luzon, Philippines (15.1N, 120.4 E). After preliminary eruptions from 12 June 1991, the climatic phase started at 05:30 UTC on 15 June 1991 and lasted for approximately 9 hours. The volcanic cloud contained gasses and particles of ice, ash, and sulfate, and reached a maximum altitude of 40 km (Holasek et al., 1996). Ice and ash burden peaked at about 80 and 50 Tg respectively, and early formed sulfate mass was estimated at 4 Tg, based on infrared satellite data from the Advanced Very High Resolution Radiometer and TIROS Operational Vertical Sounder/High Resolution Infrared Radiation Sounder/2 sensors (AVHRR, TOVS/HIRS/2; Guo et al., 2004a). Initial sulfur dioxide (SO2) mass estimates from the ultraviolet Total Ozone Mapping Spectrometer (TOMS) and infrared TOVS sensors, indicated that the eruption injected 14-22 Tg of SO2 (Bluth et al., 1992; Guo et al., 2004a). Other uncertainties pertain to the vertical extension of the volcanic cloud: SO2 mass was injected between 18-30 km (Bluth et al., 1992; Baran et al., 1993) and concentrated around 25 km, over a rich ash layer peaking around 22 km (Guo et al., 2004b). The sulfuric acid cloud peaked at 14 Tg in September (Lambert et al., 1993; Baran and Foot, 1994), with the largest aerosol concentration between 20 to 25 km of altitude and much lower amounts between 15 and 20 km (Winker and Osborn, 1992a, b; DeFoor et al., 1992). Recent volcanic SO2 emission databases suggest for Pinatubo an amount and location of SO2 emitted between 15 and 18 Tg of SO2, at an altitude of between 19 and 28 km (Independent Volcanic Eruption Source Parameter Archive Version 1.0, ivespa.co.uk, VolcanEESM: Global volcanic sulphur dioxide (SO2) emissions database from 1850 to present - Version 1.0, Multi-Decadal Sulfur Dioxide Climatology from Satellite Instruments; Aubry et al., 2021; Neely III and Schmidt, 2016; Carn, 2022).*

*Several modelling studies have evaluated the simulated global and tropical sulfate loadings compared to observations, with some studies (Niemeier et al., 2009; Toohey et al., 2011; Brühl et al., 2015) finding agreement when emitting in the mid-range of the best-estimate stratospheric SO2 loading of 14-22 Tg SO2 (Guo et al., 2004a). In contrast, a number of recent studies 50 found agreement only when injecting an amount of SO2 below the lower limit of that observed, considering different injection heights and vertical distributions (Dhomse et al., 2014; Sheng et al., 2015a; Mills et al., 2016); this difference partly motivate the design of the ISA-MIP HErSEA intercomparison (see Timmreck et al., 2018)."*

**Line 42: "are constrained across participating models": do you mean that they are the same for all participating models right? I think the language could be a bit more clear.**

Different models may have different ways of prescribing the optical parameters - "constrained" rather than "the same" helped highlight the possible differences. However, after reworking the introduction, we decided to delete this sentence.

**Line 45: "This approach…has been shown to reduce discrepancies in reproducing …anomalies". Compared to what other approach?**

Compared to free-running, fully interactive simulations. For the same reasons as the previous comment, we have deleted the sentence.

**Line 39-55: Overall I find these paragraphs a bit hard to follow. Make sure that the language is explicit for the non-expert reader, and I would suggest reorganizing them a bit: i) start by describing results of the Tambora experiment and large discrepancies between models; then highlight consequences i.e. ii) the use of a single set of aerosol optical properties derived from a simplistic model for VolMIP; and iii) the need for ISA-MIP.**

We have revised the paragraph to make it more clear. As we focus in our study on the comparison of global interactive aerosol models  we  will refer here now only to VolMIP  wrt to the Tambora study as a VolMIP pre-experiment.

**Line 54: Do you mean "lifetime" instead of "amount"? Sure different lifetime will ultimately affect the evolution of the aerosol burden, but lifetime would reflect better the characteristic affected by the effective radius.**

Yes. We have corrected this.

**Line 61: replace "initial conditions" by volcanic emission source parameters" or something like that to be more explicit?**

Corrected.

**Line 77: Why not also comparing the radiative forcing to observations? I guess this falls more under the remit of VolMIP, but it would still be of interest to many people to see which set of model/eruption source parameters result in the most realistic forcing? Radiative flux at the top of atmosphere are available from the ERBE instrument.**

This is definitely something worth doing, but we felt this paper was already very long, and decided to push this to a future paper. Additionally, as these models have been run without an interactive ocean, the comparison with ERBE (which can measure TOA imbalance) would not be straightforward.

**Line 85-87: Maybe briefly discuss what's a realistic thickness for the injected SO2 cloud? I'm not sure if we have good constraints for the Pinatubo SO2 cloud. 3D plume model simulation suggest that the thickess of the gas phase should be about**

**10%** **of** **the** **column** **height** **(see** **Figure** **S2** **in** **Aubry** **et** **al.,** **2019,**
**https://doi.org/10.1029/2019GL083975).**

We have now better acknowledged the uncertainty related to this aspect. We have added the following at the beginning of the Experimental protocol section to explain the reasons that led to these choices:

*"There is a degree of uncertainty over the thickness of the injected SO2 cloud, based on available measurements. Therefore, different modelling centers may have selected in the past different simulated injection altitudes for the Pinatubo eruption. Within (Dhomse et al., 2020) UM-UKCA set the SO2 injection altitude at 21-23 km based on the altitude of the first detection of the Pinatubo cloud at Mauna Loa (Antuña et al., 2002). Further UM-UKCA analysis by Shallcross (2020) demonstrated improved model correspondence with the July-Aug 1991 Mauna Loa lidar measurements when running the model with "prenudged free-running", rather than the "approximate QBO free-running" approach used in (Dhomse et al., 2020). Sheng et al. (2015b) performed with AER 2-D 300 atmospheric simulations of the Pinatubo eruption by varying the emission parameters and found agreement with several observations by injecting 14 Tg of SO2 with a vertical distribution peaking at 18-21 km. Similar emission parameters (10-12 Tg of SO2 at 18-20 km) were used in Mills et al. (2016) with CESM1-WACCM. Niemeier et al. (2009) showed comparable aerosol optical depth and effective radius with satellite and lidar measurements, simulating with MAECHAM5-HAM the injection of 17 Tg of SO2 at about 24 km together with 100 Tg of fine ash at about 21 km. Stenchikov et al. (2021) simulated with WRF-Chem v3.7.1 the same amounts of SO2 and ash but centred at 17 km showing that the radiative heating of ash can raise the sulfur cloud by 7 km during the first week of the eruption. These differences motivated the design of the ISA-MIP HErSEA intercomparison."*

**Line 90: Explicitly acknowledge why SO2 is injected in this way in UM-UKCA, i.e. it's already trying to fix the lack of SH transport in this model. This is a major difference in the injection set-up and UM-UKCA should be singled out on all figures/tables like EMAC (see MC2).**

Added as suggested in comment about Table 2.

**Line 91: For EMAC, either here or in the EMAC section, give more details on what these 3D-mixing ratio are in particular clarify how long after the eruption these 3D perturbation were constrained from observations (days? Weeks?), whether the injection date is modified accordingly in the model (it could affect e.g. the time at which peak SAOD is reached). Please also clarify the total mass of SO2 injected for Pinatubo and Hudson in EMAC for comparison with other experiments.**

We added the information concerning the amount and the timing of SO2 injection and the altitude of the maximum SO2 mixing ratios of the volcanic plumes for Pinatubo and Cerro Hudson, as reported in Table 2 of Schallock et al. (2021).

**Line 95: But I guess SO2 radiative effect (or ash) is not included in any of the models? It might be worth briefly acknowledging and discussing Stenchikov et al. (2021, https://doi.org/10.1029/2020JD033829)**

We added some references to radiative heating from SO2 and ash at the end of the paragraph as follow:

*"Radiative heating of ash and SO2 is also important for the initial uplift of the volcanic cloud (Lary et al., 1994; Young et al., 1994; Gerstell et al., 1995), but the contribution of SO2 is smaller than that of ash, in the first week, or sulfate aerosols, in the subsequent weeks (Stenchikov et al., 2021). About 80 Tg of ash was injected during the Pinatubo eruption (Guo et al., 2004b). However, both ash and SO2 radiative effects are not included in all model simulations as it is outside the scope of the project which focuses on the long-term evolution of the Pinatubo volcanic cloud."*

**Line 100: so only one ensemble member for ULAQ right? Make this explicit.**

Corrected.

**Section 2.1.1: You don't discuss at all the initial QBO phase. It looks like there was no attempt to pick a phase consistent with that at the time of the Pinatubo eruption (although models with nudged QBO will have this right, which isn't explicitly discussed)? This should be discussed for sure with citations of corresponding literature. How much would QBO phase affect your results in particular in terms of aerosol residence time in the tropics and SH transport?**

The QBO phase has be consistent through the post-eruption period, and this was added in the experimental protocol section. More details on the effect of the QBO is discussed later, in reference to comment at lines 534-535.

*"The evolution of the quasi-biennal oscillation (QBO) must be consistent through the post-eruption period, as it affects the dispersion of the volcanic plume to mid-latitudes (Trepte and Hitchman, 1992; Baldwin et al.; Punge et al., 2009), and consequently the size distribution and lifetime of stratospheric aerosols (Hommel et al., 2015; Pitari et al., 2016; Visioni et al., 2017). Accordingly, models with internally generated QBO re-initialized it in order to be consistent with the actual meteorological conditions, or used specified dynamics approaches (e.g. Telford et al., 2008)."*

**Line 103: I would find it clearer if you replaced "six" by "five" and in the next sentence say something like "closely related simulations from a sixth model, EMAC, are considered".**

Corrected.

**Line 119-120: Maybe try to improve consistency in terms of the order of information given across model subsections? It will make comparison easier for the reader. E.g. always have horizontal and vertical resolution after the list of models coupled, then information on QBO, then information on microphysics, etc.** Corrected.

**Line 121-122: you don't include information on how ensemble were produced for other model so be consistent? Also I'm not too familiar with this method. How long before 1991 was the rate of snow formation changed? I guess it would take some times to get really different initial states?**

We added a sentence in the experimental protocol section that refers to the sections describing the models where the generation of the ensemble for each model is explained. Moreover, for ECHAM6-SALSA we specified that *"Ensemble members were produced by using insignificantly different values for one of the tuning parameters (the rate of snow formation by aggregation) for January 1991 of each ensemble member."* The author of the simulations with ECHAM6-SALSA explains that this method allows the atmospheres of each ensemble member to build up independently several months before the eruption took place, a sufficiently short time before the eruption to have the same QBO phase.

**Line 136: Acknowledge somewhere explicitly that 4 models out of 6 have some version of ECHAM as their host model (also see MC4)**

We added this information at the end of section 2.1.2 where the participating models are introduced as follows:

*"ECHAM5-HAM, SOCOL-AERv2 and EMAC are based on the same general circulation model (GCM), ECHAM5, but with different horizontal and/or vertical resolutions, ECHAM6-SALSA has the update version ECHAM6.3, but all have different chemical and aerosol modules."*

**Line 174: Are these the same SST dataset as mentioned line 97? If so redundant info.**

No, they are two different dataset. The one used by UM-UKCA is a merged product based on the monthly mean Hadley Centre sea ice and SST dataset version 1 (HadISST1) and version 2 (Hurrell et al. 2008)

**Line 177: I obviously know nothing about author contributions in the Schallock et al. (2021) paper, but I was surprised not to see the lead author of this study among the co-authors or mentioned in the acknowledgement section given the use of the Schallock et al. (2021) simulations.**

We  will thank Jennifer Schallock in the Acknowledgement.

**Line 189: Here or where injection strategy for all models is discussed, give more details on these 3D injections.**

We added those details in the injection strategy section (as for the comment to line 91).

**Table 1: "Band" is misleading, see MC2.** Corrected.

**Section 2.2: Using the ERBE radiative flux and adding a figure comparing simulated vs observed TOA forcing would be a nice addition, even though this is more VolMIP than ISA-MIP remit.** Please see the comment in reference to line 77.

**Lines 209-215 and 221-223: Could you clarify assumptions – e.g. on aerosol size distribution – required to derive parameters describing the aerosol (surface area density, effective radius, etc) from observations of optical properties? Should "observations" for these parameters be considered equal to e.g. SAOD observations or the direct balloon measurements?**

We added more details on how the retrieved variables are calculated from observations of optical properties for SAGE II, HIRS and OPC  in section 2.2.2-4, respectively. It has been

noted in more recent papers that there might be issues related with measurements of effective radius and SAD. While Kovilakam et al., (2015) noted that the SAD in SAGE II v7.0 "is significantly better agreement and within the ± 40% precision of the OPC moment calculations.", in a personal communication with Dr. Thomason he noted that those measurements will be further improved, as shown during the 2022 SPARC assembly (https://research.reading.ac.uk/sparc-ga2022/wp-content/uploads/sites/279/2022/10/Poster Session_BS1_BS2_20Oct.pdf , N. Ernest and L. Thomason, Deriving aerosol size distributions from the University of Wyoming optical particle counter measurements at SAGE II wavelengths, poster BS1-32".

**Line 250: give resolution in degree latitude instead, and specify somewhere that GloSSAC provides zonally averaged values.** We changed the resolution in degree and added this information to line 250 and in section 2.2.5.

**Line 252 "tropical cloud core" instead of "tropical core"?** Corrected.

**Line 255-256: I don't find it that clear that Med-22 significantly overestimate SAOD for ULAQ, UKCA and EMAC?**

This can be better understood by looking at the timeseries of the SAOD averaged in the tropical region, NH and SH extratropics (Figure 1 - not included in the paper). It is more correct to say that *"overestimate the stratospheric AOD in the tropics or/and in the Northern Hemisphere (NH) extratropics compared to both observations."* In ULAQ-CCM, the SAOD of Med-22km is concentrated in a narrower band than the observation that, averaging over the tropical region (20°S - 20°N), results to be underestimated. Therefore, we leave the sentence as given above.

[Figure]

**Figure 1.** Time evolution of stratospheric AOD in the tropics, in the NH and SH midlatitudes simulated by models for Med-22km, compared with the observations.

**Line 259: don't use "band".** "Band" changed in "meridional-spread emission (0-15°N)"

**Lines 258-261: If the result that SH transport can't be reproduced holds when including Cerro Hudson (MC1), you might want to formulate more explicitly the hypothesis that point injection is not a viable option for large-magnitude eruptions?**

We include the answer in the next comment.

**Line 261: at some point in the SH transport discussion (here or later in the paper), you might want to briefly mention Jones et al. (2017, https://www.nature.com/articles/s41467-017-01606-0), especially their figure 1? For**

**the HadGEM model, it shows transport towards both hemispheres for a 23-28km injection but not for a 16-23km injection. This also motivates my comment MC2 to run point injection with UM-UKCA at different heights.**

We modified the paragraph as follow:

*"and EMAC, contrary to other models, show more southward transport, probably due to the different injection settings (see section 2.1.1). In UM-UKCA\* the meridional-spread emission (0-15N) accounts for the initial west-southwestward drift of the volcanic cloud (Bluth et al., 1992), contributing to a more hemispherically symmetric aerosol distribution (Dhomse et al., 2014; Mills et al., 2016; Jones et al., 2017). EMAC used a 3D-plume injection and also included smaller eruptions such as that of Cerro Hudson in the southern hemisphere in August 1991 (45.9S, 72.9W). The additional injection is a 3D-plume injection of 0.65 Tg-S of SO2, whose maximum in terms of mixing ratio is at 18 km, and differs from the two additional cases performed with ULAQ-CCM (2.1.1.1). In ULAQ-CCM, the Med-22km+Low-Hud includes a similar amount of SO2 but at lower altitudes compared to the Cerro Hudson eruption in EMAC, and its effect on the stratospheric burden and AOD is negligible. In contrast, Med-22km+High-Hud enhances them in the southern hemisphere, approaching observation, but only for a few months after the eruption (Fig. S6)."*

**Line 276-277: true but the SH:NH SAOD ratio also looks pretty bad for this model?**

Yes, therefore we change the sentence to

*"In ECHAM5-HAM the injection at 21-23 km results in a comparable stratospheric AOD in the tropics and SH extratropics compared to both observations, but overestimates Northern Hemispheric (NH) extratropics values by up to a factor of two."*

[Figure]

**Figure 2.** Time evolution of stratospheric AOD in the tropics, in the NH and SH midlatitudes simulated by ECHAM-HAM for Low-22km, Med-22km and High-22km, compared with the observations.

**Table 2 caption: be explicit about what correlation is considered here, and what RMSD, and also refer to appendix A1 for more details (same comment for figure 3 caption).**

Added.

**Table 2: add stars for EMAC and UM-UKCA here and in every figure/table. In captions you could say something like "\* highlight models with spatially spread SO2 injections."**

Added.

**Line 284: really too bad that there is no experiment with other heights for UM-UKCA, nor experiment with point source (MC2). A few additional experiments would take a maximum of one or two weeks to run on UK HPC systems? Marshall et al. (2019, https://doi.org/10.1029/2018JD028675) should be discussed at some point for the role of injection height in UM-UKCA**

Please see the response to comment MC2.

**Figure 2 caption: Could you discuss briefly here and/or in the main text how big of a difference is expected between SAOD/extinction between the minimum and maximum wavelength used in different models/observational dataset? Checking Pinatubo simulations with the EVA_H model (an extension of Matt Toohey's EVA), I get up to 5% differences between 525nm and 600nm for global mean SAOD. I don't think the wavelength difference would affect your results (e.g. error metric, best scenario) too much but this should be acknowledged more clearly.**

We have saved the AOD at about 550 and 1020 nm, therefore we can't provide an evaluation of the differences of SAOD between 550 and 600 nm, but we agree that the differences should be negligible. Therefore, we refer to Clyne et al. (2021) discussion of the extinction efficiency for wavelengths between 440 and 690 nm.

**Figure 3: to make this figure easier to read, maybe you could have an empty taylor diagram at the bottom right of the figure with labelled arrows showing what metric changes how when moving one direction or another on the diagram.**

We thank the reviewer for their suggestion, but after numerous attempts we decided the figure would look too messy this way and have decided not to modify it. However, considering the perplexities that may arise from Tayor diagrams, we decided to describe their results in more detail and in relation to Figure 2.

**Figure 4: Obviously important discrepancies between AVHRR and GloSSAC between month 8 and 21, but there is an apparent sudden "bump" around month 10. Could this be Cerro Hudson? (cf MC1) ECHAM6 and SOCOL capture very well the beginning and end of the AOD decrease.**

The x-axis represents the months after the SAOD has reached its peak, that is November 1991 for AVHRR, therefore the bump is around June 1992, 1 year after the eruption, when the contribution of Cerro Hudson to the SAOD is zero. Relative maxima are due to the monthly availability of the latitudes in which the measurements are taken (see Figure 3).

[Figure]

**Figure 3.** Time evolution of stratospheric AOD zonally and globally averaged (panels a and b, respectively).

**Figure 4: add star for UM-UKCA; it would be nice to have the raw global mean SAOD values provided as supplementary data (also see MC3).**

Done.

**Line 286: I'm not a fan of using this definition to calculate the e-folding time as: i) it uses a single threshold instead of capturing the full decay trend in the data; ii) it uses the SAOD instead of the total S burden, and the SAOD is affected by things like the effective radius etc (it makes more sense to fit a mass decay than a SAOD decay). On point (i) could you quickly test if your results are comparable if you instead get the e-folding times by fitting exponential decay models to the data in Figure 4 (on a linear or log scale)?**

(i) We initially calculated the e-folding time as you have suggested, but discarded this method because it was not suitable for application to observational data. We therefore decided to use the definition of e-folding time as the time for SO2 mass reduction by a factor of 1/e. (ii) We calculated the e-folding for both SAOD and S burden precisely to highlight this aspect.

**Line 328: "This might depend on the different vertical concentrations of OH in the model": be explicit on whether they increase or decrease with altitude and whether this is consistent with SO2 burden evolution.**

We deleted this sentence as we found that the reason for this difference is mainly due to the injection altitude relative to the tropopause (see also next comment) and that the discussion on OH and SO2 oxidation using monthly data is not really relevant. We changed the paragraph as follows:

*"The global normalised SO2 burden curves (Fig. S4a) coincide for all models with prescribed OH. An exception of Med-19km in ECHAM6-SALSA, which has lower values*

*and might depend on an early removal through tropopause flux, facilitated by injection near the tropopause ."*

**Line 332-334: briefly discuss how consistent these results are with observational constraint on SO2 e-folding time dependence on altitude (see Figure 14 in Carn et al. 2016, http://dx.doi.org/10.1016/j.jvolgeores.2016.01.002)**

Carn et al. (2016) show that the SO2 e-folding time increases with the increasing altitude of SO2 injection compared to the local tropopause. Accordingly, based on the monthly mean values of SO2 stratospheric burden (figure S4) we can only qualitatively say that for ECHAM6-SALSA we find SO2 e-folding time is slightly larger in Med-22km than in Med-19km that we attribute to an increase of the tropopause flux for injections closer to the tropopause. But on these lines, we emphasize the role of the vertical amplitude of injection and not the altitude with respect to the tropopause, in relation to the contribution of the OH oxidation, for which we refer to Mills et al. (2017).

**Line 341: I'm not sure why this should be the case. Sure the characteristic timescale for SO2 -> sulfate aerosol conversion is shorter than the sulfate aerosol lifetime, but there will be a more or less small fraction (depending on injection height and mass) of sulfate aerosol lost before the full mass of SO2 is converted into aerosol?**

We agree, and therefore revise the text and the next paragraph to:

*"Thus, in the build-up phase we would expect all the curves for all experiments to reach a value of 1, since no SO2 and sulfate aerosols have yet been removed from the atmosphere. This will highlight the differences in the aerosol removal (wet removal, deposition, sedimentation) depending on the injection altitude and differences in microphysical growth, especially in the descending phase. Not all models and experiments, however, reach the value of 1: ECHAM5-HAM in Med-19km and Med-18-25km, ULAQ-CCM in Med-19km, and ECHAM6-SALSA, SOCOL-AERv2 and UM-UKCA in all experiments never do. This is due to the use of monthly averages for our analyses and the faster removal, near the tropopause, of sulfate aerosol and SO2 not yet converted to aerosols, especially in Med-19km and Med-18-25km experiments."*

**Line 350: replace "by" by "with"** Corrected.

**Line 351: Here and everywhere else where you say "injection rate", replace by "injected SO2 mass". The key parameter is how much SO2 you inject, not how quickly you inject it in the models (even though this might also have an influence especially when comparing basaltic to silicic eruptions, but it's not the aim of your experimental design).** Corrected.

**Line 352: "Figure 3 shows that the differences" (that instead of comma)** Corrected.

**Line 354: do you mean 22km instead of 19 for the three scenarios?** Yes. Corrected.

**Line 365-367: please see MC1 and update the range of plausible eruption source parameters to 0.75-2Tg S and 12-18km with citation of MSVOLSO2L4 and Neely and Schmidt (2016, https://doi.org/10.5285/76ebdc0b-0eed-4f70-b89e-55e606bcd568). In IVESPA (see earlier comment), for the largest phase of the Cerro Hudson eruption,**

we have 16+/-3km for the plume top height and 17.5+/-3km for the ash injection height, with no good constraint found for the SO2 height. Added.

**Line 369: peak location of what?** The location of the stratospheric AOD peak. We make the sentence clearer.

**Line 383: you mean panel b and e instead of c and f?** Yes. Corrected.

**Line 386: "injection rate" -> correct everywhere, see previous comment** Corrected.

**Line 390: does not instead of doesn't** Corrected.

**Line 391: remove one occurrence of "especially …after the eruption"** Corrected.

**Line 390-391: Acknowledge Marshall et al. (2019) where they show that higher injection heights result in aerosol being in slower branch of the BDC and longer tropical confinement?**

In this paragraph we are discussing the sensitivity of transport to injection rates (the amount of SO2) and not heights, which is discussed right after. Hence we don't think the suggested reference (cited elsewhere already) is fitting.

**Line 396: "in which aerosols…high latitudes" -> mention that this effect is season-dependent?**

We moved this paragraph to the discussion section and add the following sentence to introduce the SAGE II observations:

*"We note that the strength of the meridional transport is also seasonally dependent, and therefore eruptions happening in other seasons would result in different distributions of the aerosol cloud (Visioni et al., 2019)."*

**Line 411: How is the mean effective radius calculated? Is it weighted by e.g. aerosol concentration? If not you might get large differences purely related to the vertical distribution of aerosols in the different datasets?**

The stratospheric effective radius is weighted by the surface area density, for the vertical profiles of reff and SAD are shown in the next figure. The calculation of both effective radius and stratospheric effective radius is specified in appendix A2. I added the reference to that appendix at that line.

**Line 418: "steady" instead of "flat"?** Corrected.

**Figure 7: replace "ratio" by "aerosol mass fraction"?.** Changed also in the whole section.

**Figure 7 g-i: Is the sum of each row not equal to 100% because of aerosol outside 60S-60N? This really confuses me. If so could you standardize wrt the mass within 60S-60N?**

We feel like it's important to show the fraction with respect to the overall burden and not just 60S-60N, to highlight both the mid-latitudinal transport but also the overall mass changes.

**Figure 7: Why is the +/-10% band highlighted in grey? Is this deemed a reasonable agreement and if so how do you justify the threshold? If no justification just have a horizontal line at 0 instead.** We removed it and included a horizontal line at 0 instead.

**Figure 7 caption: the burden (mass) is an extensive variable so it makes no sense to take its spatial average. Do you mean "total burden" instead of "global average burden"?**

We corrected the whole sentence in :

*"The aerosol mass fraction is calculated with respect to the total burden, for the tropical burden (20°N-20°S, first column, a, d, g), the burden integrated over the northern mid-latitudes (35-60°N, second column, b, e, h) and over the southern mid-latitudes (35-60°S, third column, c, f, i)."*

**Line 426: add "of ISA-MIP" after "experiment".** Corrected.

**Line 429: "since the simulated decay onset time is anticipated": I don't understand what this means, reformulate please.**

We change the sentence to:

*"After the eruption, all models are able to capture the same decay rate as the SAGE II measurements, remaining flat around the peak reached approximately after October 1991. Most produce a comparable tropical effective radius for about a couple of years, based on different injection settings."*

**Line 456-457: This refers to figure 9c? The discrepancy between observations is much smaller than the inter-model spread though?**

This refers to Figure 9c therefore I enumerated the panels in the figure and specified it in the sentence. The magnitude of the discrepancy between the observation compared to the inter-model spread depends on the altitude and period considered (see Figures 9, S4 and S5) therefore we haven't added details on this.

**Line 493: replace "mechanism" by "process"?** Corrected.

**Line 501-503: comment on how UKCA differ? While noting that the injection strategy differ.**

We can now comment on transport in UM-UKCA after the sulfate burden has been provided (we discussed the transport based on its ratio in three different regions). We made these changes:

*"However, we find a common problem in transport, either too fast from the tropics to high northern latitudes (ECHAM6-SALSA, ECHAM5-HAM, SOCOL-AERv2), confined in the NH (UM-UKCA for point injection), or too confined to the tropics (ULAQ-CCM). [...] UM-UKCA bypassed the SH transport problem by distributing the injection of SO2 between 0 and 15◦N (merdional-spread emission), also achieving a longer persistence of the volcanic aerosol cloud in the stratosphere (Figures 5 and S2, Table S2)."*

**Line 514: could or might be crucial, not would?** Changed to "could".

**Line 513: in addition to a longer lifetime it would result to slower latitudinal transport because BDC speed decreases with height? Also cite Stenchikov et al. (2021) in this paragraph.**

Paragraph changed to:

*"The lack of ash co-emission, a process not included in HErSEA simulations, could be crucial in the first days/month to better reproduce the initial cloud evolution (Mills et al., 2017; Stenchikov et al., 2021). On one hand, the ash may have removed part of the initial sulfur cloud through the SO2 or H2SO4 uptake on these coarse particles, which have a significant fall velocity (Zhu et al., 2020); on the other hand, the presence of smaller ash particles causes greater heating and vertical lofting of the volcanic cloud (Niemeier et al., 2021; Kloss et al., 2021), which could result in slower meridional l transport and longer lifetimes of stratospheric volcanic aerosols, depending on the latitude and injection altitude of SO2 (Niemeier et al., 2009; Stenchikov et al., 2021) ."*

**Line 520: At least one experiment with Cerro Hudson (MC1) would be really good to test how the lifetime is sensitive to the inclusion of this additional eruption.**

A significant contribution to the stratospheric sulfate burden is observed with the additional injection of 4 Tg of SO2 by Cerro Hudson. The additional injection increases the SAOD/burden for a few months after the eruption and does not change the e-folding time (13 months in all Med-22km simulations with ULAQ-CCM). See response to MC1 for changes in the Discussion section.

[Figure]

**Figure 4.** Time evolution of global stratospheric burden normalised to the maximum value, simulated by ULAQ-CCM for three experiments (coloured lines) and compared with observations (black lines) .

**Line 525-528: this sentence is very long and hard to follow; please rephrase and break down.**

The sentence is rephrased as follow:

*"Laakso et al. (2022), for instance, used the same climate model (ECHAM-HAMMOZ) with two different aerosol microphysics schemes, one sectional and one modal. Even just this difference produced an effective radius up to 52% greater in the sectional scheme than in the modal scheme simulation for the same amount of injected SO2."*

**Line 531: define w\* for the non-expert reader.** Corrected.

**Line 534-535: not much discussion on that, and in particular you barely discuss QBO configuration in your experiments?**

We added the following sentence:

"In our case, the experimental protocol requires the consistency of the QBO with observations through the post-eruption period; nonetheless, there are smaller scale processes and variability that are not reproducible by models with a coarse resolution that would affect the initial state of the system, as the formation of mesocyclone during the first day after the eruption (Chakraborty et al., 2009) or the passage of Typhoon Yunya within 75 km northeast the eruption (Oswalt et al., 1996)."

**Line 535: another relevant reference is Jones et al (2016, https://doi.org/10.1002/2016JD025001)** Added.

**Line 540: Do you mean 18-25km.** Corrected.

**Line 563: Also cite the recent perspective paper by Marshall et al. (2022, https://doi.org/10.1007/s00445-022-01559-3).** Added.

**References**

[revised manuscript text omitted]

---

## Author Comment (AC2)

**Reviewers' comments are in bold.** Authors' responses are in blue.

General comment

**This study uses a multi-model ensemble of global aerosol simulations performed within ISA-MIP HErSEA to assess the effect on volcanic stratospheric aerosol of uncertainties related to the $SO_2$ injection (height and amount) by the 1991 Pinatubo eruption. As a main result, the study identifies large inter-model differences as well as common limitations, particularly related to a too strong simulated meridional transport of aerosol in the northern hemisphere, that results in a faster simulated decay of the post-eruption enhancement of the stratospheric aerosol layer compared to observations. The study also highlights how different $SO_2$ injections are required for different models to "best match" observations (and how these vary for the chosen observed parameter as well).**

**I have only minor comments on the study, which I found overall well-conceived and well conducted. My evaluation of the study considers it as a "MIP" study, so based on results from a predefined protocol-driven set of experiments. I recognize that some aspects of the study remain open to discussion and thus require further investigation (the role of the Cerro Hudson and the role of ash emission as far as comparison with observations is concerned, but also the causes of the found inter-model differences). This calls for a retrospective on the HErSEA protocol (was it effective or has any weakness emerged?) and for a discussion about the implications of the findings for the original purpose of the experiment and for the purpose of ISA-MIP in general (this is mentioned for instance in lines 61-62 of the manuscript). As another general comment on the study, I encourage a more explicit discussion (if not presentation) of within-model uncertainties, intended as differences between realizations of an experiment with the same model. These might be negligible in most cases, but this is not stated and, instead, there are occasions where illustration of results from individual realizations reveals distinct behaviors (for instance in Figure 3). I have some more specific comments on this below.**

**I have also just a few minor editorial comments, as in my opinion the manuscript is overall well-structured and well written. As a general comment, I felt there is a difference in style between sections 3.1 and 3.2 (just focused on presentation of results) and section 3.2 (which mixes introduction, results and discussion, especially from the paragraph starting on line 374 onward). Maybe the authors could consider some homogenization, for instance by moving some of the more discussive parts of section 3.2 in section 4.**

We thank the reviewer for his suggestion. We moved most of the discussion of section 3.2 in the discussion section, changing much of its structure.

**Then, the manuscript could serve as a reference for future analyses based on the HErSEA experiments, especially as far as final choices in the experiment setup differ from the original protocol. In this sense, it may be worth to provide any guideline provided for the generation of the ensemble, and how this was actually done for each model. I see that for most models this is not reported, while in the other cases it is**

**not clear if the parameter perturbation was maintained for the whole simulation or just for some initial steps (ECHAM6-SALSA).**

We specified in the experimental protocol section that "The generation of the ensemble for each model is explained in the respective sections describing the model." and we did as mentioned. In particular for ECHAM6-SALSA we specified that "Ensemble members were produced by using insignificantly different values for one of the tuning parameters (the rate of snow formation by aggregation) for January 1991 of each ensemble member."

**Specific comments**

**Line 44-46: maybe it is worth mentioning here that a possible cause of the inter-model discrepancies in radiative fluxes are minor differences in forcing implementation.**

We have revised the paragraph to make it more clear. As we focus in our study on the comparison of global interactive aerosol models we will refer now only to VolMIP wrt to the Tambora study as a VolMIP pre-experiment and do not discuss VolMIP results in general.

**Line 58: proposed cooling is unclear, maybe "a certain cooling target"?**

We specified the proposed cooling target in order to be clear, as follows:

"The Geoengineering Model Intercomparison Project Phase 6 (GeoMIP6, Kravitz et al., 2015) also includes experiments with injection of stratospheric sulfate aerosols precursors in an amount to reduce the net radiative forcing from the SSP5-8.5 scenario to the SSP2-4.5 scenario "

**Line 61: to me initial conditions refer to the initial state of the system as a whole, so more than the "initial conditions of SO₂ injection" that is implicated here. I recommend the authors to always explicit this to avoid confusion. Also, other "initial conditions" such as the phase and amplitude of the QBO may be relevant here and deserve some explicit consideration in the presentation and discussion of results (see also comments below).**

We specified that initial conditions refer to the different SO2 injection settings and defined in section 2.1.1 (Experimental Protocol) the implementation of QBO, which is discussed in the results section.

**Line 161: by climatological do you mean "observed" values during the simulated period?**

We changed "climatological" in "observed".

**Line 267: is this related to the QBO phase? There seem to be little information regarding this aspect in the presentation of results and discussion. If the model spontaneously produces a QBO, it would be instructive to know how QBO phase and amplitude compare with observations. In this regard, one of the realizations of ECHAM6-SALSA is clearly different from the other two, especially in terms of rms (see Figure 3): what is the reason behind this difference? I wonder if the ensemble**

mean is truly representative for this model at least. This might motivate some focus on individual realizations as well (or on sub-ensembles).

We have now discussed in the paper the details of the experimental protocol, which prescribed a QBO consistent with observations, also for models with interactive QBO (that needed to control for consistency in their QBO state). Therefore, the observed intra-ensemble differences can't be due to different QBO states. We have added the following phrase:

*"The evolution of the quasi-biennal oscillation (QBO) must be consistent through the post-eruption period, as it affects the dispersion of the volcanic plume to mid-latitudes (Trepte and Hitchman, 1992; Baldwin et al.; Punge et al., 2009), and consequently the size distribution and lifetime of stratospheric aerosols (Hommel et al., 2015; Pitari et al., 2016; Visioni et al., 2017). Accordingly, models with internally generated QBO re-initialized it in order to be consistent with the actual meteorological conditions, or used specified dynamics approaches (e.g. Telford et al., 2008)."*

**Line 354: why not testing the differences? Even if the sample size is low, a Mann-Whitney U test, for instance, could provide you a basis for a stronger statement here.**

We prefer to show that the differences between the ensemble members of the same scenarios in ECHAM6-SALSA can be larger than the the differences between the ensemble mean of different scenarios as in Figure 1 (S1 in the supplementary material): the thick line represent the ensemble mean of each scenario and the shaded area the region between the minimum and maximum value between the ensemble members.

We added this figure in the supplementary material and referred to it in that paragraph.

[Figure]

**Figure 1.** Time evolution of the stratospheric AOD in the northern (NH) and southern hemisphere (SH) simulated by ECHAM6-SALSA, ECHAM5-HAM and SOCOL-AERv2 for the experiments with different masses of SO2 injected at about 22 km altitude. The thick

line represents the ensemble mean, the shaded area the region between the minimum and maximum value between the ensemble members (thin lines).

**Figure 8: especially for the Laramie comparison, given the punctual location of the datum, would it make sense to consider more explicitly the individual realizations instead of just the ensemble mean in order to include uncertainties linked to the "internal component" of atmospheric circulation? I understand that also due to the vertical averaging this might still lead to small differences across realizations, but it would be important to have some estimate of the uncertainty anyway (for instance an error bar at the peak value of the profile). Also, the error bar for the OPC data is not defined.**

At the beginning of section 3.3 we refer to Appendix A2 for all calculations related to the effective radius and error bar. We find that adding the shaded areas (that represent the area between the minimum and maximum values of the three ensemble members) makes the figure too messy for ECHAM6-SALSA and doesn't add any further information that has not already been discussed.

**Technical corrections**

**Line 332: typo (produces)** Corrected.

**Line 391: twice especially, maybe the second can be skipped** Corrected.

**Line 425: at analysing** Corrected.

**Line 574: typo Higher.** Corrected.

**Figure 3: I had some difficulties tracking the colors. I suggest using a more varied color palette for the different experiments.** We changed the colors using a diverging scheme ("RdYlBu") for which we made sure that was colorblind safe. The same palette is used for the comparison of experiments, with the exception of the figures where experiments of all models are compared at the same time (Figures 7, S1, S2, S7). In that case, we left the different linestyle for the experiment, as specified in the caption ("Experiments are identified here with different line styles, the different colors refer to the models.")

**References**

Baldwin, M. P., Gray, L. J., Dunkerton, T. J., Hamilton, K., Haynes, P. H., Randel, W. J., Holton, J. R., Alexander, M. J., Hirota, I., Horinouchi, T., Jones, D. B. A., Kinnersley, J. S., Marquardt, C., Sato, K., and Takahashi, M.: The quasi-biennial oscillation, Reviews of Geophysics, 39, 179–229, https://doi.org/https://doi.org/10.1029/1999RG000073, 2001.

Hommel, R., Timmreck, C., Giorgetta, M. A., and Graf, H. F.: Quasi-biennial oscillation of the tropical stratospheric aerosol layer, Atmospheric Chemistry and Physics, 15, 5557–5584, https://doi.org/10.5194/acp-15-5557-2015, 2015.

Kravitz, B., Robock, A., Tilmes, S., Boucher, O., English, J. M., Irvine, P. J., Jones, A., Lawrence, M. G., MacCracken, M., Muri, H., Moore, J. C., Niemeier, U., Phipps, S. J., Sillmann, J., Storelvmo, T., Wang, H., and Watanabe, S.: The Geoengineering Model Intercomparison Project Phase 6 (GeoMIP6): simulation design and preliminary results, Geoscientific Model Development, 8, 3379–3392, https://doi.org/10.5194/gmd-8-3379-2015, 2015.

Pitari, G., Visioni, D., Mancini, E., Cionni, I., Di Genova, G., and Gandolfi, I.: Sulfate Aerosols from Non-Explosive Volcanoes: Chemical-Radiative Effects in the Troposphere and Lower Stratosphere, Atmosphere, 7, 85, https://doi.org/10.3390/atmos7070085, 2016.

Punge, H. J., Konopka, P., Giorgetta, M. A., and Müller, R.: Effects of the quasi-biennial oscillation on low-latitude transport in the stratosphere derived from trajectory calculations, Journal of Geophysical Research: Atmospheres, 114, https://doi.org/https://doi.org/10.1029/2008JD010518, 2009.

Telford, P. J., Braesicke, P., Morgenstern, O., and Pyle, J. A.: Technical Note: Description and assessment of a nudged version of the new dynamics Unified Model, Atmospheric Chemistry and Physics, 8, 1701–1712, https://doi.org/10.5194/acp-8-1701-2008, 2008.

Trepte, C. R. and Hitchman, M. H.: Tropical stratospheric circulation deduced from satellite aerosol data, Nature, 355, 626–628, https://doi.org/10.1038/355626a0, 1992.

Visioni, D., Pitari, G., Aquila, V., Tilmes, S., Cionni, I., Di Genova, G., and Mancini, E.: Sulfate geoengineering impact on methane transport and lifetime: results from the Geoengineering Model Intercomparison Project (GeoMIP), Atmospheric Chemistry and Physics, 17, 11 209–11 226, https://doi.org/10.5194/acp-17-11209-2017, 2017.

---

## Editor Decision (ED1)

Dear Authors,

Thank you for your great efforts in addressing the reviewers' comments and revising the manuscript. After reading through your replies and revision, I have the following comments that need to be addressed before the final publication of your manuscript in ACP. The line numbers given below are those in the revision file with track changes.

1. Your reply to MC3 (copied below)

*MC3) I see no comment on data availability which is crucial before publication. In particular, having SI tables or a netcdf archive with the processed data displayed on key figures (for both model and observations) would be really welcome (at least for figures 2, 5, 8). This would facilitate comparison to your results for future studies.*

*We apologize for the lack of link - we planned to include it during the revision process as the process is not straightforward. All data is now available online and the DOI has been provided in the section "Data availability".*

I checked the DOI you provided and found the 2-D and 3-D data files from the model outputs. However, I was not able to find "*the processed data displayed on key figures (for both model and observations)*".
Please provide the processed data displayed on all figures for both model and observations (including figures in the supplement) and list them clearly in the readme file.

2. L51: You changed 7 Tg-S to 14 Tg. Is this 14 Tg for sulfuric acid mentioned at the beginning of the sentence? If yes, should be

7 Tg-S = ~7*98/32=21.4 Tg H2SO4

3. L61: You mentioned here "below the lower limit". Could you provide the exact values used in the cited studies?

4. L77: What is "volc-long-eq"? Didn't find it in other places in the text and in the replies to comments.

5. L78: Add some relevant references after "set up".

6. L87-88: Could you provide values or range of simulated aerosol burden mentioned here?

7. L131, L134: Put cited author name out of ( )?

8. L152: EMAC simulation**s**

9. L159: Please eruption time and injection amount for Spurr and Lascar mentioned here.

10. L335-336: What is "H2SO4 density"? Sulfate particle number density?

11. L345: please provide a unit for column density. Do you refer to the mass column density?

12. L417: " e " a typo?

13. L454-455: Please briefly explain how global mean SAODs were calculated, especially for AVHRR with lots of missing data (Fig 2i).

14. L566: fig. → Fig.

15. L630: Was any weight applied for the average calculation?

Best,

Fangqun

ACP co-Editor

---

## Author Response (AR2)

**Minor revisions by editor**

Dear editor, we have applied the changes you requested. We are also in the process of uploading the remaining missing data, and this might take 24-48h from the admin side to reflect on the DOI landing page. Since I (the main author) would need to have the paper published as soon as possible for graduation purposes, I hope you don't mind that we submitted this revision before the changes in the uploaded data might have happened. We assure you they are on their way. Best, Ilaria.

1. I checked the DOI you provided and found the 2-D and 3-D data files from the model outputs. However, I was not able to find "the processed data displayed on key figures (for both model and observations)". Please provide the processed data displayed on all figures for both model and observations (including figures in the supplement) and list them clearly in the readme file.

We have uploaded the processed data and explained in the ReadMe file their structure. They should be available in 1 or 2 days.

2. L51: You changed 7 Tg-S to 14 Tg. Is this 14 Tg for sulfuric acid mentioned at the beginning of the sentence? If yes, should be 7 Tg-S = ~7*98/32=21.4 Tg H2SO4

The "7 Tg-S" in the previous version was wrong, then we changed it to "14 Tg" that refers to Tg of sulfate aerosols.

3. L61: You mentioned here "below the lower limit". Could you provide the exact values used in the cited studies?

We provided the value (10 Tg of SO2).

4. L77: What is "volc-long-eq"? Didn't find it in other places in the text and in the replies to comments.

We removed the reference to the specific experiments as it is irrelevant.

5. L78: Add some relevant references after "set up".

 We added the reference to Marshall et al., (2018).

6. L87-88: Could you provide values or range of simulated aerosol burden mentioned here?

Added.

7. L131, L134: Put cited author name out of ( )?

Corrected.

8. L152: EMAC simulations.

Changed to "the EMAC simulation".

9. L159: Please eruption time and injection amount for Spurr and Lascar mentioned here.

Added.

10. L335-336: What is "H2SO4 density"? Sulfate particle number density?

Changed to "H2SO4 particle number density".

11. L345: please provide a unit for column density. Do you refer to the mass column density?

Changed to "column number density of sulfuric acid aerosols".

12. L417: " e " a typo?

Corrected.

13. L454-455: Please briefly explain how global mean SAODs were calculated, especially for AVHRR with lots of missing data (Fig 2i).

We explained this at the beginning of the section. We added a reference to the previous explanation.

14. L566: fig.  Fig.

Corrected.

15. L630: Was any weight applied for the average calculation?

We added that the stratospheric effective radius is weighted by the SAD as explained in Appendix A2 (see L627).